# SPA: A Graph Spectral Alignment Perspective for Domain Adaptation

**Zhiqing Xiao**[13], **Haobo Wang**[23], **Ying Jin**[4], **Lei Feng**[5], **Gang Chen**[13], **Fei Huang**[6], **Junbo Zhao**[13*]

[1] College of Computer Science and Technology, Zhejiang University
[2] School of Software Technology, Zhejiang University
[3] Key Lab of Intelligent Computing based Big Data of Zhejiang Province, Zhejiang University
[4] CUHK-SenseTime Joint Lab, The Chinese University of Hong Kong
[5] School of Computer Science and Engineering, Nanyang Technological University
[6] Alibaba Group

`{zhiqing.xiao, wanghaobo, cg, j.zhao}@zju.edu.cn`,
`{sherryying003, lfengqaq, feirhuang}@gmail.com`

## Abstract

Unsupervised domain adaptation (UDA) is a pivotal form in machine learning to extend the in-domain model to the distinctive target domains where the data distributions differ. Most prior works focus on capturing the inter-domain transferability but largely overlook rich intra-domain structures, which empirically results in even worse discriminability. In this work, we introduce a novel graph SPectral Alignment (SPA) framework to tackle the tradeoff. The core of our method is briefly condensed as follows: (i)-by casting the DA problem to graph primitives, SPA composes a coarse graph alignment mechanism with a novel spectral regularizer towards aligning the domain graphs in eigenspaces; (ii)-we further develop a fine-grained message propagation module — upon a novel neighbor-aware self-training mechanism — in order for enhanced discriminability in the target domain. On standardized benchmarks, the extensive experiments of SPA demonstrate that its performance has surpassed the existing cutting-edge DA methods. Coupled with dense model analysis, we conclude that our approach indeed possesses superior efficacy, robustness, discriminability, and transferability. Code and data are available at: `https://github.com/CrownX/SPA`.

## 1  Introduction

Domain adaptation (DA) problem is a widely studied area in computer vision field [20, 19, 73, 65, 40], which aims to transfer knowledge from label-rich source domains to label-scare target domains where dataset shift [32] or domain shift [72] exists. The existing unsupervised domain adaptation (UDA) methods usually explore the idea of learning domain-invariant feature representations based on the theoretical analysis [3]. These methods can be generally categorized several types, *i.e.*, moment matching methods [44, 51, 70], and adversarial learning methods [18, 66, 19, 52].

The most essential challenge of domain adaptation is how to find a suitable utilization of intra-domain information and inter-domain information to properly align target samples. More specifically, it is a trade-off between discriminating data samples of different categories within a domain to the greatest extent possible, and learning transferable features across domains with the existence of domain shift. To achieve this, adversarial learning methods implicitly mitigate the domain shift by driving the feature extractor to extract indistinguishable features and fool the domain classifier.

---

*Corresponding author.

37th Conference on Neural Information Processing Systems (NeurIPS 2023).

The adversarial DA methods have been increasingly developed and followed by a series of works [14, 45, 8]. However, it is very unexpected that the remarkable transferability of DANN [18] is enhanced at the expense of worse discriminability [10, 38].

To mitigate this problem, there is a promising line of studies that explore graph-based UDA algorithms [9, 87, 54]. The core idea is to establish correlation graphs within domains and leverage the rich topological information to connect inter-domain samples and decrease their distances. In this way, self-correlation graphs with intra-domain information are constructed, which exhibits the homophily property whereby nearby samples tend to receive similar predictions [55]. There comes the problem that is how to transfer the inter-domain information with these rich topological information. Graph matching is a direct solution to inter-domain alignment problems. Explicit graph matching methods are usually designed to find sample-to-sample mapping relations, requiring multiple matching stages for nodes and edges respectively [9], or complicated structure mapping with an attention matrix [54]. Despite the promise, we find that such a point-wise matching strategy can be restrictive and inflexible. In effect, UDA does not require an exact mapping plan from one node to another one in different domains. Its expectation is to align the entire feature space such that label information of source domains can be transferred and utilized in target domains.

In this work, we introduce a novel graph spectral alignment perspective for UDA, hierarchically solving the aforementioned problem. In a nutshell, our method encapsulates a coarse graph alignment module and a fine-grained message propagation module, jointly balancing inter-domain transferability and intra-domain discriminability. Core to our method, we propose a novel spectral regularizer that projects domain graphs into eigenspaces and aligns them based on their eigenvalues. This gives rise to coarse-grained topological structures transfer across domains but in a more intrinsic way than restrictive point-wise matching. Thereafter, we perform the fine-grained message passing in the target domain via a neighbor-aware self-training mechanism. By then, our algorithm is able to refine the transferred topological structure to produce a discriminative domain classifier.

We conduct extensive evaluations on several benchmark datasets including DomainNet, OfficeHome, Office31, and VisDA2017. The exprimental results show that our method consistently outperforms existing state-of-the-art domain adaptation methods, improving the accuracy on 8.6% on original DomainNet dataset and about 2.6% on OfficeHome dataset. Furthermore, the comprehensive model analysis demonstrates the the superiority of our method in efficacy, robustness, discriminability and transferability.

## 2 Preliminaries

**Problem Description.** Given source domain data $\mathcal{D}_s = \{(x_i^s, y_i^s)\}_{i=1}^{N_s}$ of $N_s$ labeled samples associated with $C_s$ categories from $\mathcal{X}_s \times \mathcal{Y}_s$ and target domain data $\mathcal{D}_t = \{x_i^t\}_{i=1}^{N_t}$ of $N_t$ unlabeled samples associated with $C_t$ categories from $\mathcal{X}_t$. We assume that the domains share the same feature and label space but follow different marginal data distributions, following the Covariate Shift [67]; that is, $P(\mathcal{X}_s) \neq P(\mathcal{X}_t)$ but $P(\mathcal{Y}_s \mid \mathcal{X}_s) = P(\mathcal{Y}_t \mid \mathcal{X}_t)$. Domain adaptation just occurs when the underlying distributions corresponding to the source and target domains in the shared label space are different but similar enough to make sense the transfer [15]. The goal of unsupervised domain adaptation is to predict the label $\{y_i^t\}_{i=1}^{N_t}$ in the target domain, where $y_i^t \in \mathcal{Y}_t$, and the source task is $\mathcal{X}_s \rightarrow \mathcal{Y}_s$ assumed to be the same with the target task $\mathcal{X}_t \rightarrow \mathcal{Y}_t$.

**Adversarial Domain Adaptation.** The family of adversarial domain adaptation methods, *e.g.*, Domain Adversarial Neural Network (DANN) [18] have become significantly influential in domain adaptation. The fundamental idea behind is to learn transferable features that explicitly reduce the domain shift. Similar to standard supervised classification methods, these approaches include a feature extractor $F(\cdot)$ and a category classifier $C(\cdot)$. Additionally, a domain classifier $D(\cdot)$ is trained to distinguish the source domain from the target domain and meanwhile the feature extractor $F(\cdot)$ is trained to confuse the domain classifier and learn domain-invariant features. The supervised classification loss $\mathcal{L}_{cls}$ and domain adversarial loss $\mathcal{L}_{adv}$ are presented as bellow:

$$\mathcal{L}_{cls} = \mathbb{E}_{(x_i^s, y_i^s) \sim \mathcal{D}_\mathcal{S}} \mathcal{L}_{ce}\left(C\left(F\left(x_i^s\right)\right), y_i^s\right)$$
$$\mathcal{L}_{adv} = \mathbb{E}_{F(x_i^s) \sim \tilde{\mathcal{D}}_s} \log\left[D\left(F\left(x_i^s\right)\right)\right]$$
$$+ \mathbb{E}_{F(x_i^t) \sim \tilde{\mathcal{D}}_t} \log\left[1 - D\left(F\left(x_i^t\right)\right)\right] \tag{1}$$

where $\tilde{\mathcal{D}}_s$ and $\tilde{\mathcal{D}}_t$ denote the induced feature distributions of $\mathcal{D}_s$ and $\mathcal{D}_t$ respectively, and $\mathcal{L}_{ce}(\cdot, \cdot)$ is the cross-entropy loss function.

# 3 Methodology

In this section, we will give a specific introduction to our approach. The overall pipeline is shown in Figure 1. Our method is able to effectively utilize both intra-domain and inter-domain relations simultaneously for domain adaptation tasks. Specifically, based on our constructed dynamic graphs in Section 3.1, we propose a novel framework that utilizes graph spectra to align inter-domain relations in Section 3.2, and leverages intra-domain relations via neighbor-aware propagation mechanism in Section 3.3. Our approach enables us to effectively capture the underlying domain distributions while ensuring that the learned features are transferable across domains.

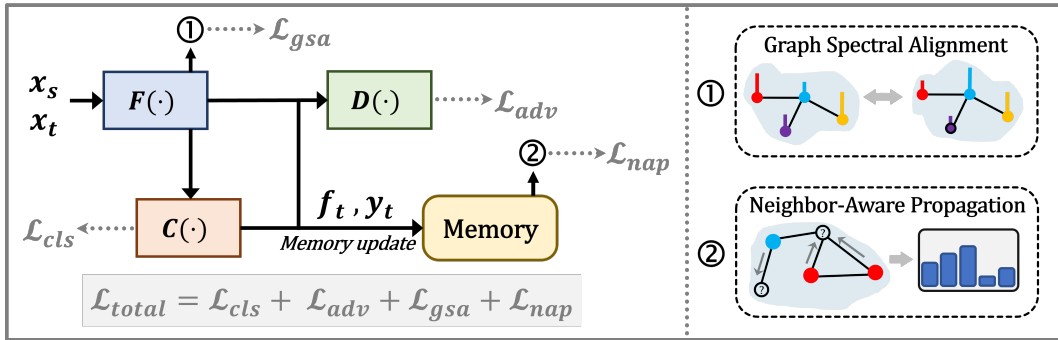

Figure 1: The overall architecture. The final objective integrates supervised classification loss $\mathcal{L}_{cls}$, domain adversarial loss $\mathcal{L}_{adv}$, neighbor-aware propagation loss $\mathcal{L}_{nap}$, and graph spectral alignment loss $\mathcal{L}_{spa}$.

## 3.1 Dynamic Graph Construction

Images and text data inherently contain rich sequential or spatial structures that can be effectively represented using graphs. By constructing graphs based on the data, both intra-domain relations and inter-domain relations can be exploited. For instance, semantic and spatial relations among detected objects in an image [43], cross-modal relations among images and sentences [11], cross-domain relations among images and texts [9, 27] have been successfully modeled using graph-based representations. In this paper, leveraging self-correlation graphs enables us to model the relations between different samples within domain and capture the underlying data distributions.

Our self-correlation graphs are constructed on source features and target features respectively. A feature extractor $F(\cdot)$ can be designed to learn source features $f_s$ and target features $f_t$ from source domain samples $\mathbf{x}_s$ and target domain samples $\mathbf{x}_t$ respectively, $i.e.$, $f_s = F(x_s)$ and $f_t = F(x_t)$. Given the extracted features $f_s$, we aim to construct a undirected and weighted graph $\mathcal{G}_s = (\mathcal{V}_s, \mathcal{E}_s)$. Each vertex $v_i \in \mathcal{V}_s$ is represented by a feature vector $f_i^s$. Each weighted edge $e_{i,j} \in \mathcal{E}_s$ can be formulated as a relation between a pair of entities $\delta(f_i^s, f_j^s)$, where $\delta(\cdot)$ denotes a metric function. With the extracted features $f_t$, another graph $\mathcal{G}_s = (\mathcal{V}_t, \mathcal{E}_t)$ can be constructed in the same way. Note that both $f_s$ and $f_t$ keep evolving along with the update of parameter $\theta$ during the training process. Their adjacency matrices are denoted as $\mathbf{A}_s$ and $\mathbf{A}_t$ respectively. As is well-known, the adjacency matrix of a graph contains all of its topological information. By representing these graphs in domain adaptation tasks, we can directly obtain the intra-domain relations and regard inter-domain alignment as a graph matching problem [9].

## 3.2 Graph Spectral Alignment

In this section, we will introduce how to align inter-domain relations for source domain graphs and target domain graphs. In domain adaptation scenarios where domain shift exits, if we directly construct a graph with source and target domain features together, we can only obtain a graph beyond the homophily assumption. In this way, a necessary domain alignment comes out.

As stated in Section 3.1, based on our correlation graphs, the inter-domain alignment can be regard as a graph matching problem. Explicit graph matching methods aim to find a one-to-one correspondence between nodes or edges of two graphs and typically involve solving a combinatorial optimization problem for node matching and edge matching respectively [9]. Nevertheless, our goal is often to align the distributions of the source and target domains and learn domain-invariant features, instead of requiring such an complicated matching approach. Therefore, we prefer a implicit graph alignment method, avoiding multiple stages of explicit graph matching.

The distance between the spectra of graphs is able to measure how far apart the spectrum of a graph with $n$ vertices can be from the spectrum of any other graph with $n$ vertices [1, 69, 22], leading to a simplification of measuring the discrepancy of graphs. Inspired by this, we give the definitions of graph laplacians and spectral distances:

**Definition 1.** (GRAPH LAPLACIANS [77]). *Let $\mathcal{G} = (\mathcal{V}, \mathcal{E})$ be a finite graph with vertices $\mathcal{V}$ and weighted edges $\mathcal{E}$. Let $\phi : \mathcal{V} \to \mathcal{R}$ be a function of the vertices taking values in a ring and $\gamma : \mathcal{E} \to \mathcal{R}$ be a weighting function of weighed edges. Then, the graph Laplacian $\Delta$ acting on $\phi$ and $\gamma$ is defined by*

$$(\Delta_\gamma \phi)(v) = \sum_{w:d(w,v)=1} \gamma_{wv}[\phi(v) - \phi(w)]$$

*where $d(w, v)$ is the graph distance between vertices $w$ and $v$, and $\gamma_{wv}$ is the weight value on the edge $wv \in \mathcal{E}$.*

**Definition 2.** (SPECTRAL DISTANCES) . *Given two simple and nonisomorphic graphs $\mathcal{G}_s$ and $\mathcal{G}_t$ on $n$ vertices with the spectra of Laplacians $\Lambda_s = \{\lambda_i^s\}_{i=1}^n$ with $\lambda_1^s \geq \lambda_2^s \geq \cdots \geq \lambda_n^s$ and $\Lambda_t = \{\lambda_i^t\}_{i=1}^n$ with $\lambda_1^t \geq \lambda_2^t \geq \cdots \geq \lambda_n^t$ respectively. Define the spectral distance between $\mathcal{G}_s$ and $\mathcal{G}_t$ as*

$$\sigma(\mathcal{G}_s, \mathcal{G}_t) = \|\Lambda_s - \Lambda_t\|_p, \quad p \geq 1$$

For a simple undirected graph with a finite number of vertices and edges, the definition 1 is just identical to the Laplacian matrix. With the adjacency matrix $\mathbf{A}_s$ of source domain graph $\mathcal{G}_s$, we can obtain its Laplacian matrix $\mathbf{L}_s$ and its Laplacian eigenvalues $\Lambda_s$. Similarly, we yield Laplacian matrix $\mathbf{L}_t$ and Laplacian eigenvalues $\Lambda_t$. Following the definition Def.2, we can calculate the spectral distances between source domain graph $\mathcal{G}_s$ and target domain $\mathcal{G}_t$, and thus the graph spectral penalty is defined as bellow:

$$\mathcal{L}_{gsa} = \sigma(\mathcal{G}_s, \mathcal{G}_t) \tag{2}$$

This spectral penalty measures the discrepancy of two graphs on spectrum space. Minimizing this penalty decreases the distance of source domain graphs and target domain graphs. It can also be regarded as a regularizer and easy to combine with existing domain adaptation methods.

**More Insights.** Graph Laplacian filters are a type of signal processing filter to apply a smoothing operation to the signal on a graph by taking advantage of the local neighborhood structure of the graph represented by the Laplacian matrix [6]. A graph filter can be denoted as $(f * g)_{\mathcal{G}} = U g_\Lambda U^T f$, where $f$ is a signal on graph $\mathcal{G}$, and $\Lambda$ and $U$ is the eigenvalues and eigenvectors of Laplacian matrix $L = U \Lambda U^T$. This filer is also the foundation of classic graph neural networks [7, 16, 36] and usually infers the labels of unlabeled nodes with the information from the labeled ones in a graph. Similar to the hypothesis in [91], source features and target features will be aligned into the same eigenspace along with the learning process and finally only differs slightly in the eigenvalues $g_\Lambda$.

### 3.3 Neighbor-aware Propagation Mechanism

In this section, we will introduce how to exploit intra-domain relations within target domain graphs. The well-trained source domain naturally forms tight clusters in the latent space. After aligning via the aforementioned graph spectral penalty, the rich topological information is coarsely transferred to the target domain. To perform the fine-grained intra-domain alignment, we take a further step by encouraging message propagation within the target domain graph.

Intuitively, we adopt the weighted $k$-Nearest-Neighbor (KNN) classification algorithm [83] to generate pseudo-label for target domain graph. We focus on unlabelled target samples $\{x_i^t\}_{i=1}^{N_t}$ and thus omit the domain subscript $t$ for clarity. Let $p_i = C(F(x_i))$ denotes the $C_t$-dimensional prediction of unlabelled target domain sample $x_i$ and $p_i^m$ denotes its mapped probability stored in the memory

bank. A vote is obtained from the top $k$ nearest labelled samples for each unlabelled sample $x_i \in \mathbf{x}$, which is denoted as $\mathcal{N}_i$. The vote of each neighbor $j \in \mathcal{N}_i$ is weighted by the corresponding predicted probabilities $p_{j,c}$ that data sample $x_j$ will be identified as category $c$. Therefore, the voted probability $p_{i,c}$ of data sample $x_i$ corresponding to class $c$ can be defined as $q_{i,c} = \sum_{j \neq i, j \in \mathcal{N}_i} p_{j,c}^m$. and we yield the normalized probability $\hat{q}_{i,c} = q_{i,c} / \sum_{m=1}^{C_t} q_{i,m}$ and the pseudo-label $\hat{y}$ for data sample $x_i$ can be calculated as $\hat{y}_i = \arg\max_c \hat{q}_{i,c}$. Considering different neighborhoods $\mathcal{N}_i$ lie in different local density, we should expect a larger weight for the target data in higher local density [48]. Obviously, a larger $\hat{q}_{i,c}$ means the data sample $x_i$ lies in a neighborhood of higher density. In this way, we directly utilize the category-normalized probability $\hat{q}_{i,c}$ as the confidence value for each pseudo-label and thus the weighted cross-entropy loss based on the pseudo-labels is formulated as:

$$\mathcal{L}_{nap} = -\alpha \cdot \frac{1}{N_t} \sum_{i=1}^{N_t} \hat{q}_{i,\hat{y}_i} \log p_{i,\hat{y}_i} \qquad (3)$$

where $\alpha$ is the coefficient term properly designed to grow along with iterations to mitigate the noises in the pseudo-labels at early iterations and avoid the error accumulation [39]. Note that our neighbor-aware propagation loss only depends on unlabeled target data samples, just following the classic self-training method [31].

**Memory Bank.** We design the memory bank to store prediction probabilities and their associated feature vectors mapped by their target data indices. Before storing them, we firstly apply the sharpening technique to fix the ambiguity in the predictions of target domain data [5, 24]:

$$\tilde{p}_{j,c} = p_{j,c}^{-\tau} / \sum_{x=1}^{C_t} p_{j,x}^{-\tau} \qquad (4)$$

where $\tau$ is the temperature to scale the prediction probabilities. As $\tau \to 0$, the probability will collapse to a point mass [39]. Then, we utilize L2-norm to normalize feature vectors $f_i$. Finally, we store the sharpened prediction $\tilde{p}_i$ and its associated normalized feature $\|f_i\|$ to the memory bank via $\beta$-exponential moving averaging (EMA) strategy, updating them for each iteration. The predictions stored in the memory bank $p_i^m$ are utilized to approximate real-time probability for pseudo-label generation, and the feature vectors $f_i^m$ stored in the memory bank are employed to construct graphs before neighbor-aware propagation.

**The Final Objective.** At the end of our approach, let us integrate all of these losses together, *i.e*, supervised classification loss $\mathcal{L}_{cls}$ and domain adversarial loss $\mathcal{L}_{adv}$ descried in Eq.1, the loss of neighbor-aware propagation mechanism $\mathcal{L}_{nap}$ in Eq.3, and the loss of graph spectral alignment $\mathcal{L}_{gsa}$ in Eq.2. Finally, we can obtain the final objective as follows:

$$\mathcal{L}_{total} = \mathcal{L}_{cls} + \mathcal{L}_{adv} + \mathcal{L}_{gsa} + \mathcal{L}_{nap} \qquad (5)$$

For the sake of simplicity, we leave out the loss scale for $\mathcal{L}_{adv}$ and $\mathcal{L}_{gsa}$ here. Just as the coefficient term $\alpha$ in $\mathcal{L}_{nap}$, these two losses are also adjusted with specific scales in the implementation. For more detailed implementation specifics, please refer to our source code. Note that we only describe the problem of unsupervised domain adaptation here our approach can easily to extend to semi-supervised domain adaptation scenario. Concerning the labeled data, we employ the standard cross-entropy loss with label-smoothing regularization [71] in our implementation. We also give different trials of similarity metrics and graph laplacians. More concrete details are in the following section.

## 4 Experiments

In this section, we present our main empirical results to show the effectiveness of our method. To evaluate the effectiveness of our architecture, we conduct comprehensive experiments under unsupervised domain adaptation, semi-supervised domain adaptation settings. The results of other compared methods are directly reported from the original papers. More experiments can be found in the Appendix.

### 4.1 Experimental Setups

**Datasets.** We conduct experiments on 4 benchmark datasets: 1) **Office31** [63] is a widely-used benchmark for visual DA. It contains 4,652 images of 31 office environment categories from three

domains: *Amazon* (A), *DSLR* (D), and *Webcam* (W), which correspond to online website, digital SLR camera and web camera images respectively. 2) **OfficeHome** [76] is a challenging dataset that consists of images of everyday objects from four different domains: *Artistic* (A), *Clipart* (C), *Product* (P), and *Real-World* (R). Each domain contains 65 object categories in office and home environments, amounting to 15,500 images around. Following the typical settings [10], we evaluate methods on one-source to one-target domain adaptation scenario, resulting in 12 adaptation cases in total. 3) **VisDA2017** [58] is a large-scale benckmark that attempts to bridge the significant synthetic-to-real domain gap with over 280,000 images across 12 categories. The source domain has 152,397 synthetic images generated by rendering from 3D models. The target domain has 55,388 real object images collected from *Microsoft COCO* [49]. Following the typical settings [56], we evaluate methods on synthetic-to-real task and present test accuracy for each category. 4) **DomainNet** [57] is a large-scale dataset containing about 600,000 images across 345 categories, which span 6 domains with large domain gap: *Clipart* (C), *Infograph* (I), *Painting* (P), *Quickdraw* (Q), *Real* (R), and *Sketch* (S). Following the settings in [28], we compare various methods for 12 tasks among C, P, R, S domains on the original DomainNet dataset.

**Implementation details.** We use PyTorch and tllib toolbox [28] to implement our method and fine-tune ResNet pre-trained on ImageNet [25, 26]. Following the standard protocols for unsupervised domain adaptation in previous methods [48, 56], we use the same backbone networks for fair comparisons. For Office31 and OfficeHome dataset, we use ResNet-50 as the backbone network. For VisDA2017 and DomainNet dataset, we use ResNet-101 as the backbone network. We adopt mini-batch stochastic gradient descent (SGD) with a momentum of 0.9, a weight decay of 0.005, and an initial learning rate of 0.01, following the same learning rate schedule in [48].

## 4.2 Result Comparisons

In this section, we compare SPA with various state-of-the-art methods for unsupervised domain adaptation (UDA) scenario, and *Source Only* in UDA task means the model trained only using labeled source data. We also extend SPA to semi-suerpervised domain adaptation (SSDA) scenario and conduct experiments on 1-shot and 3-shots setting. With the page limits, see supplementary material for details.

Table 1: Classification Accuracy (%) on DomainNet for unsupervised domain adaptation (inductive), using ResNet101 as backbone. The best accuracy is indicated in **bold** and the second best one is underlined. Note that we compare methods on the original DomainNet dataset with train/test splits in target dataset, leading to an inductive scenario.

| Method | C→P | C→R | C→S | P→C | P→R | P→S | R→C | R→P | R→S | S→C | S→P | S→R | Avg. |
|---|---|---|---|---|---|---|---|---|---|---|---|---|---|
| *Source Only* [25] | 32.7 | 50.6 | 39.4 | 41.1 | 56.8 | 35.0 | 48.6 | 48.8 | 36.1 | 49.0 | 34.8 | 46.1 | 43.3 |
| DAN [51] | 38.8 | 55.2 | 43.9 | 45.9 | 59.0 | 40.8 | 50.8 | 49.8 | 38.9 | 56.1 | 45.9 | 55.5 | 48.4 |
| DANN [18] | 37.9 | 54.3 | 44.4 | 41.7 | 55.6 | 36.8 | 50.7 | 50.8 | 40.1 | 55.0 | 45.0 | 54.5 | 47.2 |
| BCDM [45] | 38.5 | 53.2 | 43.9 | 42.5 | 54.5 | 38.5 | 51.9 | 51.2 | 40.6 | 53.7 | 46.0 | 53.4 | 47.3 |
| MCD [66] | 37.5 | 52.9 | 44.0 | 44.6 | 54.5 | 41.6 | 52.0 | 51.5 | 39.7 | 55.5 | 44.6 | 52.0 | 47.5 |
| ADDA [73] | 38.4 | 54.1 | 44.1 | 43.5 | 56.7 | 39.2 | 52.8 | 51.3 | 40.9 | 55.0 | 45.4 | 54.5 | 48.0 |
| CDAN [52] | 39.9 | 55.6 | 45.9 | 44.8 | 57.4 | 40.7 | 56.3 | 52.5 | 44.2 | 55.1 | 43.1 | 53.2 | 49.1 |
| MCC [31] | 40.1 | 56.5 | 44.9 | 46.9 | 57.7 | 41.4 | 56.0 | 53.7 | 40.6 | 58.2 | 45.1 | 55.9 | 49.7 |
| JAN [53] | 40.5 | 56.7 | 45.1 | 47.2 | 59.9 | 43.0 | 54.2 | 52.6 | 41.9 | 56.6 | 46.2 | 55.5 | 50.0 |
| MDD [42] | 42.9 | 59.5 | 47.5 | 48.6 | 59.4 | 42.6 | 58.3 | 53.7 | 46.2 | 58.7 | 46.5 | 57.7 | 51.8 |
| SDAT [62] | 41.5 | 57.5 | 47.2 | 47.5 | 58.0 | 41.8 | 56.7 | 53.6 | 43.9 | 58.7 | 48.1 | 57.1 | 51.0 |
| Leco [80] | 44.1 | 55.3 | 48.5 | 49.4 | 57.5 | 45.5 | 58.8 | 55.4 | 46.8 | 61.3 | 51.1 | 57.7 | 52.6 |
| SPA (Ours) | **54.3** | **70.9** | **56.1** | **59.3** | **71.5** | **51.8** | **64.6** | **59.6** | **52.1** | **66.0** | **57.4** | **70.6** | **61.2** |

Table 1 (DomainNet) show the results on DomainNet dataset. We compare SPA with the various state-of-the-art methods for inductive UDA scenario on this large-scale dataset. The inductive setting refers that we use the train/test splits of target domain data as the original dataset [57]. During the training process, we can use the train split of target domain data and then we use the test split of target domain data for testing. This setting is relatively more difficult than others with the large-scale data amount and inductive learning. Only a few of works follows this setting while we give a result comparison in this setting here. The experiments shows SPA consistently ourperforms than various of DA methods with the average accuracy of 61.2%, which is 8.6% higher than the recent work Leco [80], demonstrating the superiority of SPA in this inductive setting.

Table 2: Classification Accuracy (%) on OfficeHome for unsupervised domain adaptation, using ResNet50 as backbone. The best accuracy is indicated in **bold** and the second best one is underlined.

| Method | A→C | A→P | A→R | C→A | C→P | C→R | P→A | P→C | P→R | R→A | R→C | R→P | Avg. |
|---|---|---|---|---|---|---|---|---|---|---|---|---|---|
| *Source Only* [25] | 34.9 | 50.0 | 58.0 | 37.4 | 41.9 | 46.2 | 38.5 | 31.2 | 60.4 | 53.9 | 41.2 | 59.9 | 46.1 |
| DANN [18] | 45.6 | 59.3 | 70.1 | 47.0 | 58.5 | 60.9 | 46.1 | 43.7 | 68.5 | 63.2 | 51.8 | 76.8 | 57.6 |
| CDAN [52] | 50.7 | 70.6 | 76.0 | 57.6 | 70.0 | 70.0 | 57.4 | 50.9 | 77.3 | 70.9 | 56.7 | 81.6 | 65.8 |
| BSP [10] | 52.0 | 68.6 | 76.1 | 58.0 | 70.3 | 70.2 | 58.6 | 50.2 | 77.6 | 72.2 | 59.3 | 81.9 | 66.3 |
| NPL [39] | 54.1 | 74.1 | 78.4 | 63.3 | 72.8 | 74.0 | 61.7 | 51.0 | 78.9 | 71.9 | 56.6 | 81.9 | 68.2 |
| GVB [14] | 57.0 | 74.7 | 79.8 | 64.6 | 74.1 | 74.6 | 65.2 | 55.1 | 81.0 | 74.6 | 59.7 | 84.3 | 70.4 |
| MCC [31] | 56.3 | 77.3 | 80.3 | 67.0 | 77.1 | 77.0 | 66.2 | 55.1 | 81.2 | 73.5 | 57.4 | 84.1 | 71.0 |
| BNM [13] | 56.7 | 77.5 | 81.0 | 67.3 | 76.3 | 77.1 | 65.3 | 55.1 | 82.0 | 73.6 | 57.0 | 84.3 | 71.1 |
| MetaAlign [82] | 59.3 | 76.0 | 80.2 | 65.7 | 74.7 | 75.1 | 65.7 | 56.5 | 81.6 | 74.1 | 61.1 | 85.2 | 71.3 |
| ATDOC [48] | 58.3 | 78.8 | 82.3 | 69.4 | 78.2 | 78.2 | 67.1 | 56.0 | 82.7 | 72.0 | 58.2 | 85.5 | 72.2 |
| FixBi [56] | 58.1 | 77.3 | 80.4 | 67.7 | 79.5 | 78.1 | 65.8 | 57.9 | 81.7 | 76.4 | 62.9 | 86.7 | 72.7 |
| SDAT [62] | 58.2 | 77.1 | 82.2 | 66.3 | 77.6 | 76.8 | 63.3 | 57.0 | 82.2 | 74.9 | **64.7** | 86.0 | 72.2 |
| NWD [8] | 58.1 | 79.6 | 83.7 | 67.7 | 77.9 | 78.7 | 66.8 | 56.0 | 81.9 | 73.9 | 60.9 | 86.1 | 72.6 |
| SPA (Ours) | **60.4** | **79.7** | **84.5** | **73.6** | **81.3** | **82.1** | **72.2** | **58.0** | **85.2** | **77.4** | 61.0 | **88.1** | **75.3** |

Table 3: Classification Accuracy (%) on (a) Office31 and (b) VisDA2017 for unsupervised domain adaptation, using ResNet50 and ResNet101 as backbone respectively. For VisDA2017, all the results are based on ResNet101 except those with mark [†], which are based on ResNet50. The best accuracy is indicated in **bold** and the second best one is underlined. See supplementary material for more details.

(a) Office31

| Method | A→D | A→W | D→A | D→W | W→A | W→D | Avg. |
|---|---|---|---|---|---|---|---|
| *Source Only* [25] | 78.3 | 70.4 | 57.3 | 93.4 | 61.5 | 98.1 | 76.5 |
| DANN [18] | 79.7 | 82.0 | 68.2 | 96.9 | 67.4 | 99.1 | 82.2 |
| CDAN [52] | 92.9 | 94.1 | 71.0 | 98.6 | 69.3 | 100. | 87.7 |
| MixMatch [5] | 88.5 | 84.6 | 63.3 | 96.1 | 65.0 | 99.6 | 75.4 |
| BSP [10] | 93.0 | 93.3 | 73.6 | 98.2 | 72.6 | 100. | 88.5 |
| NPL [39] | 88.7 | 89.1 | 65.8 | 98.1 | 66.6 | 99.6 | 84.7 |
| GVB [14] | 95.0 | 94.8 | 73.4 | 98.7 | 73.7 | 100. | 89.3 |
| MCC [31] | 92.1 | 94.0 | 74.9 | 98.5 | 75.3 | 100. | 89.1 |
| BNM [13] | 92.2 | 94.0 | 74.9 | 98.5 | 75.3 | 100. | 89.2 |
| MetaAlign [82] | 93.0 | 94.5 | 73.6 | 98.6 | 75.0 | 100. | 89.2 |
| ATDOC [48] | **95.4** | 94.6 | 77.5 | 98.1 | 77.0 | 99.7 | 90.4 |
| FixBi [56] | 95.0 | 96.1 | **78.7** | **99.3** | **79.4** | 100. | **91.4** |
| NWD [8] | **95.4** | 95.2 | 76.4 | 99.1 | 76.5 | 100. | 90.4 |
| SPA (Ours) | 95.0 | **97.2** | 78.0 | 99.0 | **79.4** | 99.8 | **91.4** |

(b) VisDA2017

| Method | Avg. |
|---|---|
| *Source Only* [25] | 52.4 |
| DANN [18] | 57.4 |
| CDAN [52] | 73.7 |
| MixMatch [5] | 70.4 |
| BSP [10] | 75.9 |
| NPL [39] | 74.7 |
| GVB [14] | 75.3[†] |
| MCC [31] | 78.8 |
| BNM [13] | 79.5 |
| ATDOC [48] | 86.3 |
| FixBi [56] | 87.2 |
| SDAT [62] | 82.1 |
| NWD [8] | 83.7 |
| SPA (Ours) | **87.7** |

Table 2, Table 3a, and Table 3b show the results on OfficeHome, Office31, and VisDA2017 datasets respectively. We compare SPA with the various state-of-the-art methods for transductive UDA scenario on these three public benchmarks. Table 2 (OfficeHome) shows the average accuracy of SPA is 75.3%, achieving the best accuracy, which is 2.6% higher than the second highest method FixBi [56] and 2.7% higher than the recent work NWD [8]. Note that the average accuracy reported by NWD is based on MCC [31] and SPA is based on DANN [18]. Table 3a (Office31) shows the results on the small-sized Office31 dataset, the average accuracy of SPA is 91.4%, which outperforms most of domain adaptation methods and comparable with FixBi [56]. SPA also shows a significant performance improvement over the baseline DANN. Table 3b (VisDA2017) shows the results on VisDA2017. This is a large-scale dataset with only 12 classes. SPA based on DANN achieves the accuracy of 83.7%, which have already been superior than lots of methods. However, for fair comparisons, we follow the setting of ATDOC [48] and report the accuracy based on MixMatch [5], which achieve the best accuracy of 87.7%, 0.5% better than FixBi, and 1.4% better than ATDOC.

## 4.3 Model Analysis

**Ablation Study.** To verify the effectiveness of different components of SPA, we design several variants and compare their classification accuracy. As shown in the first section in Table 4, the w/ $\mathcal{L}_{nap}$, $\mathcal{L}_{gsa}$ method is based on CDAN [52]. The difference between these variants is whether it utilize the loss of pseudo-labelling and graph spectral penalty or not. The results illustrate that both

Table 4: Classification Accuracy (%) on OfficeHome for unsupervised domain adaptation. The table is divided into three sections corresponding to the three analysis of ablation study, robustness analysis, and parameter sensitivity, each separated by a double horizontal line. More studies on other datasets are in supplementary material.

| Method | A→C | A→P | A→R | C→A | C→P | C→R | P→A | P→C | P→R | R→A | R→C | R→P | Avg. |
|---|---|---|---|---|---|---|---|---|---|---|---|---|---|
| w/o $\mathcal{L}_{gsa}, \mathcal{L}_{nap}$ | 54.6 | 74.1 | 78.1 | 63.0 | 72.2 | 74.1 | 61.6 | 52.3 | 79.1 | 72.3 | 57.3 | 82.8 | 68.5 |
| w/o $\mathcal{L}_{gsa}$ | 59.0 | 80.1 | 81.5 | 66.2 | 77.8 | 76.7 | 69.2 | 57.7 | 83.0 | 74.0 | 64.1 | 85.8 | 72.9 |
| w/o $\mathcal{L}_{nap}$ | 59.0 | 78.2 | 81.0 | 63.5 | 76.5 | 76.2 | 64.7 | 57.3 | 82.1 | 73.6 | 61.4 | 85.7 | 71.6 |
| w/ $\mathcal{L}_{gsa}, \mathcal{L}_{nap}$ | 59.9 | 79.1 | 84.4 | 74.9 | 79.1 | 81.9 | 72.4 | 58.4 | 84.9 | 77.9 | 61.2 | 87.7 | 75.1 |
| $\beta = 0.1$ | 58.3 | 79.2 | 83.2 | 72.8 | 79.3 | 80.4 | 73.3 | 58.2 | 84.5 | 79.0 | 61.4 | 87.3 | 74.7 |
| $\beta = 0.3$ | 59.0 | 79.5 | 83.8 | 73.6 | 80.6 | 81.7 | 73.5 | 58.0 | 84.9 | 78.2 | 61.3 | 87.7 | 75.1 |
| $\beta = 0.5$ | 59.3 | 79.5 | 84.1 | 73.3 | 80.6 | 81.7 | 73.1 | 58.0 | 84.9 | 78.2 | 62.2 | 87.4 | 75.2 |
| $\beta = 0.7$ | 59.9 | 79.4 | 84.0 | 73.6 | 80.5 | 82.0 | 72.7 | 57.3 | 84.8 | 77.9 | 61.9 | 87.3 | 75.1 |
| $\beta = 0.9$ | 59.7 | 79.6 | 84.7 | 72.9 | 78.7 | 82.2 | 71.2 | 57.5 | 84.5 | 77.2 | 61.7 | 87.5 | 74.8 |
| $\mathbf{L}_{rwk}$ w/ $cos$ | 60.6 | 79.4 | 84.1 | 72.5 | 79.2 | 81.8 | 71.9 | 57.1 | 84.8 | 76.3 | 61.8 | 87.4 | 74.7 |
| $\mathbf{L}_{rwk}$ w/ $gauss$ | 60.4 | 79.7 | 84.5 | 73.6 | 81.3 | 82.1 | 72.2 | 58.0 | 85.2 | 77.4 | 61 | 88.1 | 75.3 |
| $\mathbf{L}_{sym}$ w/ $cos$ | 60.7 | 79.5 | 83.8 | 72.7 | 80.0 | 81.8 | 71.5 | 57.2 | 84.8 | 76.3 | 61.9 | 87.4 | 74.8 |
| $\mathbf{L}_{sym}$ w/ $gauss$ | 59.4 | 79.5 | 84.5 | 73.6 | 81.0 | 81.7 | 72.2 | 57.6 | 84.8 | 77.5 | 61.8 | 87.8 | 75.1 |
| $\mathbf{L}_{rwk}$ w/ $k = 3$ | 59.0 | 80.6 | 83.9 | 72.4 | 79.6 | 81.7 | 71.5 | 56.5 | 84.7 | 76.4 | 62.0 | 87.7 | 74.7 |
| $\mathbf{L}_{rwk}$ w/ $k = 5$ | 60.4 | 79.7 | 84.5 | 73.6 | 81.3 | 82.1 | 72.2 | 58.0 | 85.2 | 77.4 | 61 | 88.1 | 75.3 |
| $\mathbf{L}_{sym}$ w/ $k = 3$ | 59.2 | 80.1 | 84.0 | 72.6 | 81.6 | 81.5 | 71.2 | 57.5 | 84.7 | 76.3 | 61.5 | 87.4 | 74.8 |
| $\mathbf{L}_{sym}$ w/ $k = 5$ | 59.4 | 79.5 | 84.5 | 73.6 | 81.0 | 81.7 | 72.2 | 57.6 | 84.8 | 77.5 | 61.8 | 87.8 | 75.1 |

the loss of pseudo-labelling and graph spectral penalty can improve the performance of baseline, and further combining these two losses together yields better outcomes, which is comparable with cutting-edge methods.

**Parameter Sensitivity.** To analyze the stability of SPA, we design experiments on the hyperparameter $\beta$ of exponential moving averaging strategy for memory updates. The experimental results of $\beta = 0.1$, 0.3, 0.5, 0.7, 0.9 is shown in the second section of Table 4. These results are based on CDAN [52]. From the series of results, we can find that in OfficeHome dataset, the choice of $\beta = 0.5$ outperforms than others. In addition, the differences between these results are within 0.5%, which means that SPA is insensitive to this hyperparameter.

**Robustness Analysis.** To further identify the robustness of SPA to different graph structures, we conduct experiments on different types of Laplacian matrix, similarity metric of graph relations, and $k$ number of nearest neighbors of KNN classification methods. Here are the two types of commonly-used Laplacian matrix [77]: the random walk laplacian matrix $\mathbf{L}_{rwk} = \mathbf{D}^{-1}\mathbf{A}$, and the symmetrically normalized Laplacian matrix $\mathbf{L}_{sym} = \mathbf{I} - \mathbf{D}^{-1/2}\mathbf{A}\mathbf{D}^{-1/2}$, where $\mathbf{D}$ denotes the degree matrix based on the adjacency matrix $\mathbf{A}$. In addition, the similarity metric are chosen from cosine similarity and Gaussian similarity, and different $k = 3, 5$ when applying KNN classification algorithm. The results are shown in the second section of Table 4. We can find that different types of Laplacian matrix still lead to comparable results. As for the similarity metric, the Gaussian similarity brings better performance than cosine similarity and the results also presents that 5-NN graphs is superior than 3-NN graphs in OfficeHome dataset. For all these aforementioned experiments results, the differences between them are within 1% around, confirming the robustness of SPA.

**Feature Visualization.** To demonstrate the learning ability of SPA, we visualize the features of DANN [18], BSP [10], NPL [39] and SPA with the t-SNE embedding [17] under the C → R setting of OfficeHome Dataset. BSP is a classic domain adaptation method which also use matrix decomposition and NPL is a classic pseudo-labeling method which also use high-confidence predictions on unlabeled data as true labels. Thus, we choose them as baselines to compare. The results are shown in Figure 2. Comparing Figure 2d with Figure 2a, Figure 2b and Figure 2c respectively, we make some important observations: (i)-we find that the clusters of features in Figure 2d are more compact, or say less diffuse than others, which suggests that the features learned by SPA are able to attain more desirable discriminative property; (ii)-we observe that red markers are more closer and overlapping with blue markers in Figure 2d, which suggests that the source features and target features learned by SPA are transferred better. These observations imply the superiority of SPA over discriminability and transferability in unsupervised domain adaptation scenario.

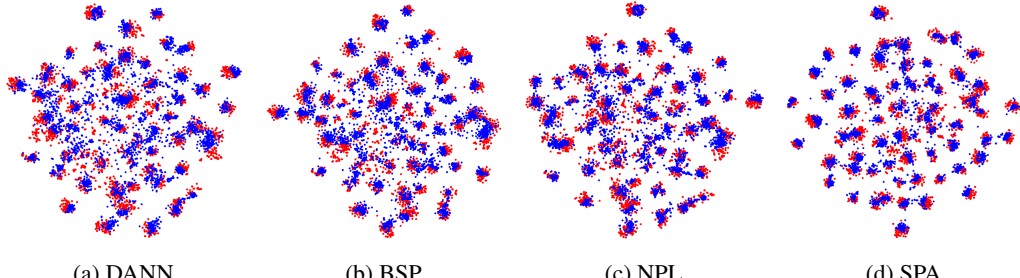

| (a) DANN | (b) BSP | (c) NPL | (d) SPA |

Figure 2: Feature Visualization. the t-SNE plot of DANN [18], BSP [10], NPL [39], and SPA features on OfficeHome dataset in the C → R setting. We use red markers for source domain features and blue markers for target domain features.

## 5 Related Work

**Domain Adaptation.** Domain adaptation (DA) problem is a specific subject of transfer learning [90, 29], and also a widely studied area in computer vision field [18, 20, 19, 73, 52, 65, 66, 40, 44, 70, 51], By the label amount of target domain, domain adaptation can be divided into unsupervised domain adaptation, and semi-supervised domain adaptation *etc.* Unsupervised domain adaptation methods are very effective at aligning feature distributions of source and target domains without any target supervision but perform poorly when even a few labeled examples are available in the target. Our approach mainly focus on unsupervised domain adaptation, achieving the cutting-edge performance. It can also be extended to semi-supervised domain adaptation with comparable results.

There are more specifics of our baselines. Inspired by random walk [77] in graph theory, MCC [31] proposes a self-training method by optimizing the class confusion matrix. BNM [13] formulates the loss of nuclear norm, *i.e*, the sum of singular values based on the prediction outputs. NWD [8] calculates the difference between the nuclear norm of source prediction matrix and target prediction matrix. The aforementioned three methods all focus on the prediction outputs and establish class-to-class relations. Besides, there are two works focusing on the features, CORAL [70], measuring the difference of covariance, and BSP [10], directly performing singular value decomposition on features and calculate top $k$ singular values. BNM, NWD and BSP are similar to our method with the use of singular values. However, our eigenvalues comes out from the graph laplacians, which contains graph topology information [12]. For example, the second-smallest non-zero laplacian eigenvalue is called as the algebraic connectivity, and its magnitude just reflects how connected graph is [12]. As we know, we are the first to propose a graph spectral alignment perspective for domain adaptation scenarios.

**Self-training.** Self-training is a semi-supervised learning method that enhances supervised models by generating pseudo-labels for unlabeled data based on model predictions, which has been explored in lots of works [27, 39, 50, 83, 84, 68]. These methods pick up the class with the maximum predicted probability as true labels each time the weights are updated. With filtering strategies and iterative approaches employed to improve the quality of pseudo labels, this self-training technique has also been applied to some domain adaptation methods [48, 88, 81, 37]. This kind of methods proves beneficial when the labeled data is limited. However, it relies on an assumption of the unlabeled data following the same distribution with the labeled data and requires accurate initial model predictions for precise results [59, 2]. The intra-domain alignment in our approach helps the neighbor-aware self-training mechanism generate more precise labels. The interaction between components of our approach brings our impressive results in DA scenarios.

**Graph Data Mining.** Graphs are widely applied in real-world applications to model pairwise interactions between objects in numerous domains such as biology [78], social media [60] and finance [86]. Because of their rich value, graph data mining has long been an important research direction [7, 16, 36, 75]. It also plays a significant role in computer vision tasks such as image retrieval [34], object detection [46], and image classification [74]. Bruna et al.[7] first introduce the graph convolution operation based on spectral graph theory. The graph filter and the eigen-decomposition of the graph Laplacian matrix inspire us to propose our graph spectral alignment.

# 6 Conclusion

In this paper, we present a novel spectral alignment perspective for balancing inter-domain transferability and intra-domain discriminability in UDA. We first leveraged graph structure to model the topological information of domain graphs. Next, we aligned domain graphs in their eigenspaces and propagated neighborhood information to generate pseudo-labels. The comprehensive model analysis demonstrates the superiority of our method. The current method for constructing graph spectra is only a starting point and may be inadequate for more difficult scenarios such as universal domain adaptation. Additionally, this method is currently limited to visual classification tasks , and more sophisticated and generic methods to object detection or semantic segmentation are expected in the future. Furthermore, we believe that graph data mining methods and the well-formulated properties of graph spectra should have more discussion in both domain adaptation and computer vision field.

# 7 Acknowledgement

This work is majorly supported in part by the National Key Research and Development Program of China (No. 2022YFB3304100), the NSFC under Grants (No. 62206247), and by the Fundamental Research Funds for the Central Universities (No. 226-2022-00028). JZ also thanks the sponsorship by CAAI-Huawei Open Fund.

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

## A  More Setups

**Hardware and Software Configurations.**  All experiments are conducted on a server with the following configurations:

- Operating System: Ubuntu 20.04.4 LTS
- CPU: Intel(R) Xeon(R) Platinum 8358P CPU @ 2.60GHz, 32 cores, 128 processors
- GPU: NVIDIA GeForce RTX 3090

**More Implementation Details.**  We use PyTorch and tllib toolbox [28] to implement our method and fine-tune ResNet pre-trained on ImageNet [25, 26]. Following the standard protocols for unsupervised domain adaptation in previous methods [48, 56], we use the same backbone networks for fair comparisons. For Office31 and OfficeHome dataset, we use ResNet-50 as the backbone network. For VisDA2017 and DomainNet dataset, we use ResNet-101 as the backbone network. Following previous work [48], we adopt mini-batch stochastic gradient descent (SGD) to learn the feature encoder by fine-tuning from the ImageNet pre-trained model with the learning rate 0.001, and new layers, as bottleneck layer and classification layer. The learning rates of the layers trained from scratch are set to be 0.01. We use the the same learning rate schedule in [48, 52], including a learning rate scheduler with a momentum of 0.9, a weight decay of 0.005, the bottleneck size of 256, and batch size of 32.

We report main experimental results with the average accuracy over 5 random trials with the initial seed 0. For transductive unsupervised domain adaptation, the reported accuracy is computed on the complete unlabeled target data, following established protocol for UDA [18, 52, 31, 10, 48]. For inductive unsupervised domain adaptation on DomainNet, the reported accuracy is computed on the provided test dataset.. We use a standard batch size of 32 for both source and target in all experiments and for all variants of our method. The reverse validation [47, 89] is conducted to select hyper-parameters. For both unsupervised domain adaptation (UDA) and semi-supervised domain adaptation (SSDA) scenarios, we fix the coefficient of $\mathcal{L}_{nap}$ as 0.2 and the coefficient of $\mathcal{L}_{gsa}$ as 1.0, while we will offer a sensitivity analysis for this two coefficients in the following section. More details refer to our code in the supplemental materials.

## B  More Experiments

### B.1  Unsupervised Domain Adaptation

In the main paper, we present the classification accuracy results on VisDA2017 dataset for unsupervised domain adaptation and leave out per-category accuracy details. In the appendix, Table 5, we give the full table on VisDA2017, using ResNet101 as backbone. Looking into this table, we can find that our SPA model consistently outperforms most of domain adaptation methods. For classic baselines, we improve DANN [18] by 30.3%, and CDAN [52] by 14 %. For recent and state-of-the-art baselines, our results is 4% higher than NWD [8] and 0.5 % better than FixBi [56]. Our SPA model ranks top in 6 out of 12 categories, ranks top 2 in 9 out of 12 categories, and SPA also achieves the best classification accuracy in total.

Furthermore, in the main paper, we present the classification accuracy results on original DomainNet with 365 categories. While the original DomainNet dataset has noisy labels, the previous work [64] use a subset of it that contains 126 categories from C, P, R and S, 4 domains in total, which we refer to as DomainNet126. In the appendix, Table 6, we show the results on DomainNet126. Our SPA model consistently ranks top among 12 tasks across 4 domains and achieves the best accuracy of 77.1 %, which is 12.3 % better than the second one. For classic baselines, we improve DANN [18] by 20.2 %, and CDAN [52] by 16 %.

### B.2  Semi-supervised Domain Adaptation

We also extend our SPA model to semi-superivsed domain adaptation (SSDA) scenario and conduct experiments on 1-shot and 3-shot setting, and $S + T$ in SSDA task means the model trained only by the labeled source and target data.

Table 5: Per-category Accuracy (%) on VisDA2017 for unsupervised domain adaptation, using ResNet101 as backbone. All the results are based on ResNet101 except those with mark $^\dagger$, which are based on ResNet50. The best accuracy is indicated in **bold** and the second best one is underlined.

| Method | aero | bike | bus | car | horse | knife | motor | person | plant | skate | train | truck | Avg. |
|---|---|---|---|---|---|---|---|---|---|---|---|---|---|
| *Source Only* [25] | 55.1 | 53.3 | 61.9 | 59.1 | 80.6 | 17.9 | 79.7 | 31.2 | 81.0 | 26.5 | 73.5 | 8.5 | 52.4 |
| DANN [18] | 81.9 | 77.7 | 82.8 | 44.3 | 81.2 | 29.5 | 65.1 | 28.6 | 51.9 | 54.6 | 82.8 | 7.8 | 57.4 |
| CDAN [52] | 85.2 | 66.9 | 83.0 | 50.8 | 84.2 | 74.9 | 88.1 | 74.5 | 83.4 | 76.0 | 81.9 | 38.0 | 73.7 |
| MixMatch [5] | 93.9 | 71.8 | 93.5 | 82.1 | 95.3 | 0.7 | 90.8 | 38.1 | 94.5 | 96.0 | 86.3 | 2.2 | 70.4 |
| BSP [10] | 92.4 | 61.0 | 81.0 | 57.5 | 89.0 | 80.6 | 90.1 | 77.0 | 84.2 | 77.9 | 82.1 | 38.4 | 75.9 |
| NPL [39] | 90.9 | 74.6 | 73.2 | 55.8 | 89.6 | 64.6 | 86.8 | 68.7 | 90.7 | 64.8 | 89.5 | 47.7 | 74.7 |
| GVB [14] | - | - | - | - | - | - | - | - | - | - | - | - | 75.3$^\dagger$ |
| MCC [31] | 92.2 | 82.9 | 76.8 | 66.6 | 90.9 | 78.5 | 87.9 | 73.8 | 90.1 | 76.1 | 87.1 | 41.0 | 78.8 |
| BNM [13] | 93.6 | 68.3 | 78.9 | 70.3 | 91.1 | 82.8 | 93.0 | 78.7 | 90.9 | 76.5 | 89.1 | 40.9 | 79.5 |
| ATDOC [48] | 95.3 | 84.7 | 82.4 | 75.6 | 95.8 | 97.7 | 88.7 | 76.6 | 94.0 | 91.7 | 91.5 | 61.9 | 86.3 |
| FixBi [56] | 96.1 | 87.8 | 90.5 | 90.3 | 96.8 | 95.3 | 92.8 | 88.7 | 97.2 | 94.2 | 90.9 | 25.7 | 87.2 |
| SDAT [62] | 94.8 | 77.1 | 82.8 | 60.9 | 92.3 | 95.2 | 91.7 | 79.9 | 89.9 | 91.2 | 88.5 | 41.2 | 82.1 |
| NWD [8] | 96.1 | 82.7 | 76.8 | 71.4 | 92.5 | 96.8 | 88.2 | 81.3 | 92.2 | 88.7 | 84.1 | 53.7 | 83.7 |
| SPA (Ours) | **98.5** | **92.2** | 86.3 | 63.0 | **97.5** | 95.4 | **93.5** | 80.7 | **97.2** | **95.2** | 91.1 | 61.4 | **87.7** |

Table 6: Classification Accuracy (%) on DomainNet126 for unsupervised domain adaptation, using ResNet101 as backbone. The best accuracy is indicated in **bold** and the second best one is underlined.

| Method | C→P | C→R | C→S | P→C | P→R | P→S | R→C | R→P | R→S | S→C | S→P | S→R | Avg. |
|---|---|---|---|---|---|---|---|---|---|---|---|---|---|
| *Source Only* [25] | 38.4 | 50.9 | 43.9 | 50.3 | 66.7 | 39.9 | 54.6 | 57.9 | 43.7 | 52.5 | 43.5 | 48.3 | 49.2 |
| DANN [18] | 46.5 | 58.2 | 51.6 | 52.7 | 64.2 | 52.9 | 61.7 | 60.3 | 53.9 | 62.7 | 56.7 | 61.6 | 56.9 |
| MCD [66] | 43.7 | 55.7 | 47.6 | 51.9 | 67.8 | 45.0 | 52.9 | 57.3 | 40.4 | 56.3 | 50.8 | 56.8 | 52.3 |
| BSP [10] | 45.7 | 58.7 | 55.5 | 48.6 | 65.2 | 48.6 | 55.2 | 60.8 | 48.6 | 56.8 | 55.8 | 61.4 | 55.1 |
| CDAN [52] | 50.9 | 61.6 | 54.8 | 59.4 | 68.5 | 55.5 | 70.4 | 66.9 | 57.7 | 64.2 | 59.1 | 64.3 | 61.1 |
| SAFN [85] | 50.0 | 58.7 | 52.4 | 56.3 | 73.7 | 53.5 | 55.8 | 64.8 | 48.5 | 60.7 | 59.5 | 64.3 | 58.2 |
| RSDA [23] | 45.5 | 56.6 | 46.6 | 45.7 | 60.4 | 48.6 | 54.6 | 61.5 | 50.9 | 56.1 | 54.0 | 58.6 | 53.4 |
| PAN [79] | 58.8 | 65.2 | 54.6 | 57.5 | 70.5 | 53.1 | 67.6 | 66.7 | 55.9 | 64.4 | 60.2 | 66.6 | 61.8 |
| MemSAC [33] | 53.6 | 66.5 | 58.8 | 63.2 | 71.2 | 58.1 | 73.2 | 70.5 | 61.5 | 68.8 | 64.1 | 67.6 | 64.8 |
| SPA (Ours) | **73.5** | **84.0** | **70.6** | **76.5** | **85.9** | **71.9** | **76.6** | **77.0** | **69.8** | **78.3** | **76.8** | **83.9** | **77.1** |

Table 7: Classification Accuracy (%) on DomainNet126 for 1-shot and 3-shot semi-supervised domain adaptation, using ResNet34 as backbone. The best accuracy is indicated in **bold** and the second best one is underlined.

| Method | C→S | | P→C | | P→R | | R→C | | R→P | | R→S | | S→P | | Avg. | |
|---|---|---|---|---|---|---|---|---|---|---|---|---|---|---|---|---|
| | 1- | 3- | 1- | 3- | 1- | 3- | 1- | 3- | 1- | 3- | 1- | 3- | 1- | 3- | 1- | 3- |
| *S + T* [25] | 54.8 | 57.9 | 59.2 | 63.0 | 73.7 | 75.6 | 61.2 | 63.9 | 64.5 | 66.3 | 52.0 | 56.0 | 60.4 | 62.2 | 60.8 | 63.6 |
| DANN [18] | 52.8 | 55.4 | 70.3 | 72.2 | 56.3 | 59.6 | 58.2 | 59.8 | 61.4 | 62.8 | 52.2 | 54.9 | 57.4 | 59.9 | 58.4 | 60.7 |
| ENT [21] | 54.6 | 60.0 | 65.4 | 71.1 | 75.0 | 78.6 | 65.2 | 71.0 | 65.9 | 69.2 | 52.1 | 61.1 | 59.7 | 62.1 | 62.6 | 67.6 |
| MME [64] | 56.3 | 61.8 | 69.0 | 71.7 | 76.1 | 78.5 | 70.0 | 72.2 | 67.7 | 69.7 | 61.0 | 61.9 | 64.8 | 66.8 | 66.4 | 68.9 |
| APE [35] | 56.7 | 63.1 | 72.9 | **76.7** | 76.6 | 79.4 | 70.4 | 76.6 | 70.8 | 72.1 | 63.0 | 67.8 | 64.5 | 66.1 | 67.6 | 71.7 |
| BiAT [30] | 57.9 | 61.5 | 71.6 | 74.6 | 77.0 | 78.6 | 73.0 | 74.9 | 68.0 | 68.8 | 58.5 | 62.1 | 63.9 | 67.5 | 67.1 | 69.7 |
| MixMatch [5] | 59.3 | 62.7 | 66.7 | 68.7 | 74.8 | 78.8 | 69.4 | 72.6 | 67.8 | 68.8 | 62.5 | 65.6 | 66.3 | 67.1 | 66.7 | 69.2 |
| NPL [39] | 62.5 | 64.5 | 67.6 | 70.7 | 78.3 | 79.3 | 70.9 | 72.9 | 69.2 | 70.7 | 62.0 | 64.8 | 67.0 | 68.6 | 68.2 | 70.2 |
| BNM [13] | 58.4 | 62.6 | 69.4 | 72.7 | 77.0 | 79.5 | 69.8 | 73.7 | 69.8 | 71.2 | 61.4 | 65.1 | 64.1 | 67.6 | 67.1 | 70.3 |
| MCC [31] | 56.8 | 60.5 | 62.8 | 66.5 | 75.3 | 76.5 | 65.5 | 67.2 | 66.9 | 68.1 | 57.6 | 59.8 | 63.4 | 65.0 | 64.0 | 66.2 |
| ATDOC [48] | 65.6 | 66.7 | 72.8 | 74.2 | **81.2** | 81.2 | 74.9 | 76.9 | 71.3 | 72.5 | 65.2 | 64.6 | 68.7 | 70.8 | 71.4 | 72.4 |
| AESL [61] | 59.5 | 65.2 | 74.6 | 76.2 | 72.4 | 74.1 | 74.8 | **77.3** | **79.4** | **80.5** | 66.2 | 69.5 | 66.6 | 69.6 | 70.5 | **73.2** |
| SPA (Ours) | **65.9** | **67.0** | **74.8** | 76.5 | 81.1 | **82.3** | **75.3** | 76.0 | 71.8 | 72.2 | 65.8 | 67.2 | **69.8** | **71.1** | **72.1** | **73.2** |

We present the classificaition accuracy results on DomainNet126 and OfficeHome datasets for SSDA scenario in the Table 7 and Table 8 respectively. Looking at the details, Table 7 shows the classification results for 1-shot and 3-shot SSDA setting on DomainNet126 dataset. For the 1-shot setting, our SPA model can improve DANN [18] by 13.7 % and ENT [21] by 9.5 %. our SPA consistently ranks top among 4 out of 7 tasks and ranks top 2 among all tasks, achieving the best accuracy of 72.1 %, which is better 1.6 % then the second one. For the 3-shot setting, our SPA model can improve DANN [18] by 12.5 % and ENT [21] by 5.6 %. SPA achieves the best accuracy of 73.2 %, comparable with recent work AESL [61].

Furthermore, Table 8 shows the classification results for 1-shot and 3-shot SSDA setting on Office-Home dataset. To verify that our SPA model can also generalize to SSDA scenario, we compare SPA with several classic and recent baselines. The first section of the table shows our SPA model can

Table 8: Classification Accuracy (%) on OfficeHome for 1-shot and 3-shot semi-supervised domain adaptation, using ResNet34 as backbone. The best accuracy is indicated in **bold** and the second best one is underlined.

| Method (1-shot) | A→C | A→P | A→R | C→A | C→P | C→R | P→A | P→C | P→R | R→A | R→C | R→P | Avg. |
|---|---|---|---|---|---|---|---|---|---|---|---|---|---|
| S + T [25] | 52.1 | 78.6 | 66.2 | 74.4 | 48.3 | 57.2 | 69.8 | 50.9 | 73.8 | 70.0 | 56.3 | 68.1 | 63.8 |
| DANN [18] | 53.1 | 74.8 | 64.5 | 68.4 | 51.9 | 55.7 | 67.9 | 52.3 | 73.9 | 69.2 | 54.1 | 66.8 | 62.7 |
| ENT [21] | 53.6 | 81.9 | 70.4 | 79.9 | 51.9 | 63.0 | 75.0 | 52.9 | 76.7 | 73.2 | 63.2 | 73.6 | 67.9 |
| MME [64] | 61.9 | 82.8 | 71.2 | 79.2 | 57.4 | 64.7 | 75.5 | 59.6 | 77.8 | 74.8 | **65.7** | 74.5 | 70.4 |
| APE [35] | 60.7 | 81.6 | 72.5 | 78.6 | 58.3 | 63.6 | **76.1** | 53.9 | 72.3 | 72.3 | 63.6 | 69.8 | 68.9 |
| CDAC [41] | 61.9 | **83.1** | 72.7 | **80.0** | 59.3 | 64.6 | 75.9 | **61.2** | 78.5 | 75.3 | 64.5 | 75.1 | 71.0 |
| SPA (Ours) | **62.3** | 76.7 | **79.0** | 66.6 | **77.3** | **76.4** | 65.7 | 59.1 | **80.7** | 71.4 | 65.2 | **84.1** | **72.0** |

| Method (3-shot) | A→C | A→P | A→R | C→A | C→P | C→R | P→A | P→C | P→R | R→A | R→C | R→P | Avg. |
|---|---|---|---|---|---|---|---|---|---|---|---|---|---|
| S + T [25] | 55.7 | 80.8 | 67.8 | 73.1 | 53.8 | 63.5 | 73.1 | 54.0 | 74.2 | 68.3 | 57.6 | 72.3 | 66.2 |
| DANN [18] | 57.3 | 75.5 | 65.2 | 69.2 | 51.8 | 56.6 | 68.3 | 54.7 | 73.8 | 67.1 | 55.1 | 67.5 | 63.5 |
| ENT [21] | 62.6 | 85.7 | 70.2 | 79.9 | 60.5 | 63.9 | 79.5 | 61.3 | 79.1 | 76.4 | 64.7 | 79.1 | 71.9 |
| MME [64] | 64.6 | 85.5 | 71.3 | 80.1 | 64.6 | 65.5 | 79.0 | 63.6 | 79.7 | 76.6 | 67.2 | 79.3 | 73.1 |
| APE [35] | 66.4 | **86.2** | 73.4 | **82.0** | 65.2 | 66.1 | **81.1** | 63.9 | 80.2 | 76.8 | 66.6 | 79.9 | 74.0 |
| CDAC [41] | **67.8** | 85.6 | 72.2 | 81.9 | 67.0 | 67.5 | 80.3 | **65.9** | 80.6 | **80.2** | **67.4** | 81.4 | 74.2 |
| SPA (Ours) | 63.1 | 81.0 | **80.2** | 68.5 | **81.7** | **77.5** | 69.5 | 65.2 | **82.0** | 73.9 | 67.2 | **87.0** | **74.7** |

improve DANN [18] by 9.3 % and ENT [21] by 4.1 % in the 1-shot setting. The second section shows our SPA model can improve DANN [18] by 11.2 % and ENT [21] by 2.8 % in the 3-shot setting. This shows that our SPA model can greatly improve the classic baselines and achieve comparable results with CDAC [41].

## B.3    Model Analysis

**Robustness Analysis.**    In the main paper, we have already verified the robustness of SPA to different graph structures on OfficeHome dataset. In the appendix, we further show the experimental results on Office31 dataset in Table 9. Similarly, we conduct experiments on different types of Laplacian matrix, similarity metric of graph relations, and $k$ number of nearest neighbors of KNN classification methods. The Laplacian matrices are chosen from the random walk laplacian matrix $\mathbf{L}_{rwk} = \mathbf{D}^{-1}\mathbf{A}$, and the symmetrically normalized Laplacian matrix $\mathbf{L}_{sym} = \mathbf{I} - \mathbf{D}^{-1/2}\mathbf{A}\mathbf{D}^{-1/2}$, where $\mathbf{D}$ denotes the degree matrix based on the adjacency matrix $\mathbf{A}$. In addition, the similarity metrics are chosen from cosine similarity and Gaussian similarity, and different $k = 3, 5$ when applying KNN classification algorithm. From Table 9, We can find that different types of Laplacian matrix still lead to comparable results. As for the similarity metric, the Gaussian similarity brings better performance than cosine similarity. On Office31 dataset, 5-NN graphs is superior than 3-NN graphs when combining with $\mathbf{L}_{rwk}$, and comparable when combining with $\mathbf{L}_{sym}$. For all these aforementioned experiments results, the differences between them are within 1% around, confirming the robustness of SPA.

Table 9: Classification Accuracy (%) on OfficeHome for unsupervised domain adaptation. The table is divided into three sections corresponding to robustness analysis, and parameter sensitivity, each separated by a double horizontal line.

| Method | A→D | A→W | D→A | D→W | W→A | W→D | Avg. |
|---|---|---|---|---|---|---|---|
| $\beta = 0.1$ | 94.2 | 95.1 | 76.6 | 98.9 | 78.6 | 99.6 | 90.5 |
| $\beta = 0.3$ | 94.0 | 96.2 | 76.9 | 98.9 | 79.3 | 99.8 | 90.8 |
| $\beta = 0.5$ | 94.0 | 96.4 | 77.9 | 99.0 | 78.0 | 99.8 | 90.8 |
| $\beta = 0.7$ | 94.4 | 96.0 | 79.0 | 98.6 | 80.3 | 100. | 91.4 |
| $\beta = 0.8$ | 94.2 | 95.7 | 76.1 | 98.6 | 78.3 | 99.8 | 90.5 |
| $\mathbf{L}_{rwk}$ w/ $cos$ | 93.8 | 95.0 | 76.3 | 98.6 | 79.7 | 100. | 90.6 |
| $\mathbf{L}_{rwk}$ w/ $gauss$ | 95.0 | 95.6 | 78.2 | 98.6 | 80.0 | 99.8 | 91.2 |
| $\mathbf{L}_{sym}$ w/ $cos$ | 93.8 | 93.8 | 78.5 | 98.6 | 79.9 | 100. | 90.8 |
| $\mathbf{L}_{sym}$ w/ $gauss$ | 93.8 | 95.6 | 79.4 | 98.6 | 78.8 | 99.8 | 91.0 |
| $\mathbf{L}_{rwk}$ w/ $k = 3$ | 93.8 | 95.2 | 78.3 | 98.9 | 80.4 | 99.8 | 91.1 |
| $\mathbf{L}_{rwk}$ w/ $k = 5$ | 93.8 | 95.0 | 76.3 | 98.6 | 79.7 | 100. | 90.6 |
| $\mathbf{L}_{sym}$ w/ $k = 3$ | 94.0 | 95.8 | 75.2 | 99.0 | 80.5 | 100. | 90.7 |
| $\mathbf{L}_{sym}$ w/ $k = 5$ | 93.8 | 93.8 | 78.5 | 98.6 | 79.9 | 100. | 90.8 |

**Parameter Sensitivity.** In the main paper, we have already analyzed the experiments on the hyperparameter $\beta$ of exponential moving averaging strategy for memory updates and verify that SPA is insensitive to this hyperparameter. Here, we show the experimental results on Office31 dataset in Table 9. Similarly, we design experiments on the hyperparameter $\beta$ of exponential moving averaging strategy for memory updates, choosing $\beta = 0.1, 0.3, 0.5, 0.7, 0.9$ respectively. These results are based on DANN [18]. From the series of results, we can find that in Office31 dataset, the choice of $\beta = 0.7$ outperforms than others. In addition, the differences between these results are within 1.0%, which means that SPA is relatively stable to this hyperparameter.

Furthermore, we design experiments for the coefficient of $\mathcal{L}_{nap}$ and the coefficient of $\mathcal{L}_{gsa}$ to analyze the stability of SPA. The experimental results are shown in the Figure 3. These results are based on DANN [18]. Fixing the coefficient of $\mathcal{L}_{nap} = 0.2$, the coefficient of $\mathcal{L}_{gsa}$ changes from 0.1 to 0.9. Fixing the coefficient of $\mathcal{L}_{gsa} = 1.0$, the coefficient of $\mathcal{L}_{nap}$ changes from 0.1 to 0.9. From the series of results, we can find that in OfficeHome dataset, the choice of different coefficients result in similar results, which means that SPA is insensitive to these coefficients.

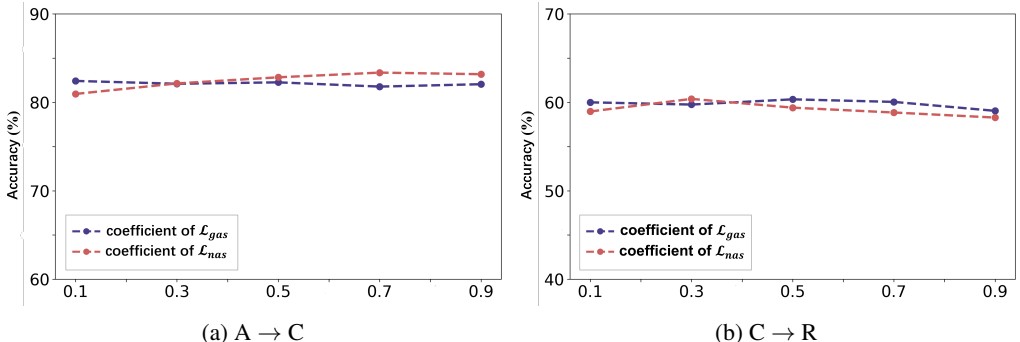

Figure 3: Parameter Sensitivity. The line plot of the coefficient of $\mathcal{L}_{nap}$ and the coefficient of $\mathcal{L}_{gsa}$ change from 0.1 to 0.9, leading to different accuracy results. The experiments are conducted on OfficeHome dataset, where (a) is A $\rightarrow$ C setting and (b) is C $\rightarrow$ R setting.

**Transferability and Discriminability.** The $\mathcal{A}$-distance [4] measures the distribution discrepancy that is defined as $d_{\mathcal{A}} = 2(1 - 2\epsilon)$, where $\epsilon$ is the classifier loss to discriminate the souce and target domains. Smaller $\mathcal{A}$-distance indicates better domain-invariant features. Figure 4a shows that SPA can achieve a lower $d_{\mathcal{A}}$, implying a lower generalization error. Furthermore, following previous work [10], we further offer the source accuracy and target accuracy specifically, in Figure 4b and Figure 4c. We can find that various methods achieve similar results of source accuracy, and SPA can always achieve higher target accuracy. Combined with the experimental results in Figure 4a, this reveals that SPA enhances transferability while still keep a strong discriminability. Back to our Introduction, this means that SPA can find a more suitable utilization of intra-domain information and inter-domain information to properly align target samples.

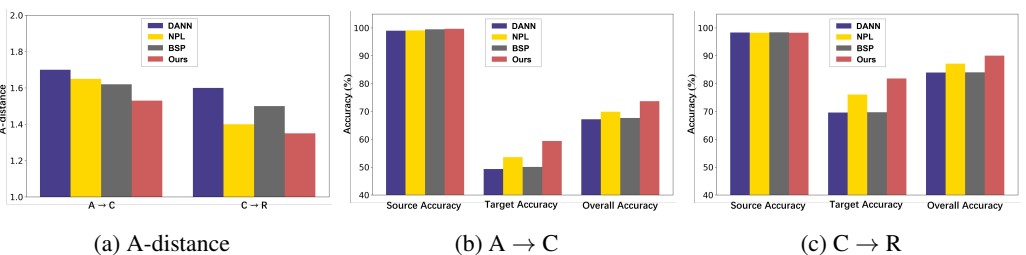

Figure 4: Transferability and Discriminability. We compare SPA with DANN [18], BSP [10] and NPL [39] on OfficeHome dataset, where (a) is $\mathcal{A}$-distance in A $\rightarrow$ C and C $\rightarrow$ R setting, (b) is accuracy results in A $\rightarrow$ C setting and (c) is accuracy results in C $\rightarrow$ R setting.

**Feature Visualization.** To demonstrate the learning ability of SPA, we visualize the features of DANN [18], BSP [13], NPL [39] and SPA with the t-SNE embedding [17] under the C → R setting of OfficeHome Dataset in the main paper, referred as Figure 2. In the appendix, we offer more visualization figures, including Figure 5 in A → D setting and Figure 6 in A → W setting on Office31 dataset, Figure 7 in A → C setting on OfficeHome dataset. According to all the figures, the observations are consistent that the source features and target features learned by SPA are transferred better. These observations imply the superiority of SPA over discriminability and transferability in unsupervised domain adaptation scenario.

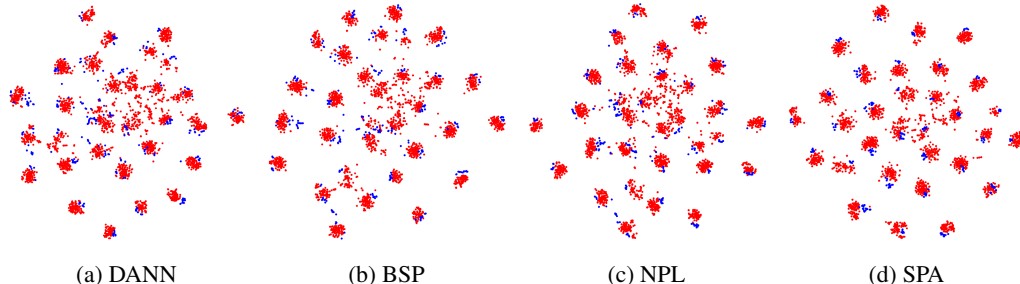

|          (a) DANN          |          (b) BSP          |          (c) NPL          |          (d) SPA          |

Figure 5: Feature Visualization. the t-SNE plot of DANN [18], BSP [13], NPL [39], and SPA features on office31 dataset in the A → D setting. We use red markers for source domain features and blue markers for target domain features.

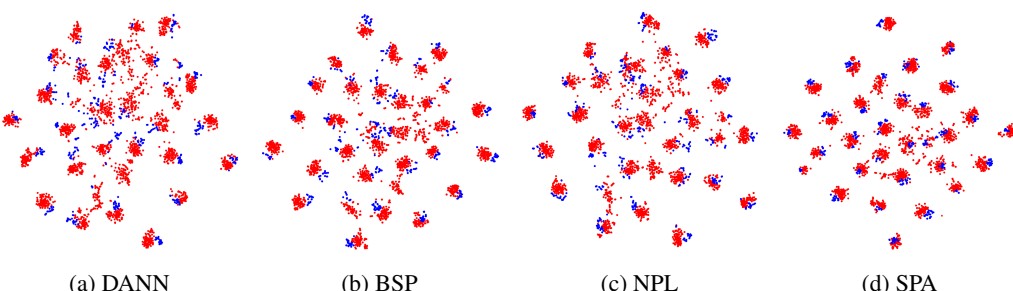

|          (a) DANN          |          (b) BSP          |          (c) NPL          |          (d) SPA          |

Figure 6: Feature Visualization. the t-SNE plot of DANN [18], BSP [13], NPL [39], and SPA features on Office31 dataset in the A → W setting. We use red markers for source domain features and blue markers for target domain features.

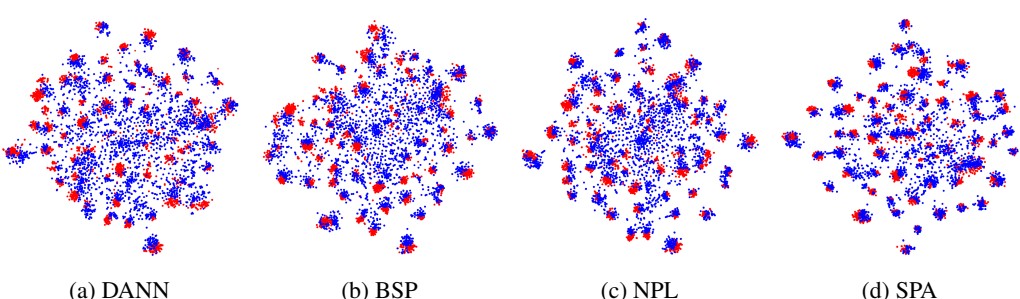

|          (a) DANN          |          (b) BSP          |          (c) NPL          |          (d) SPA          |

Figure 7: Feature Visualization. the t-SNE plot of DANN [18], BSP [13], NPL [39], and SPA features on OfficeHome dataset in the A → C setting. We use red markers for source domain features and blue markers for target domain features.

**Convergence Analysis.**    To demonstrate the convergence of SPA, we visualize the changes of accuracy and training loss respectively during the training process. As shown in Figure 8, the red curve is the changes of accuracy and the blue curve is the changes of training loss. The experiments are conducted on OfficeHome dataset, where 8a and 8b are in A → P setting, 8c and 8d are in R → A setting. Based on our experiments, it is evident that the model demonstrates satisfactory convergence properties. Through multiple iterations and varied initial settings, the model consistently approached and reached a stable state,

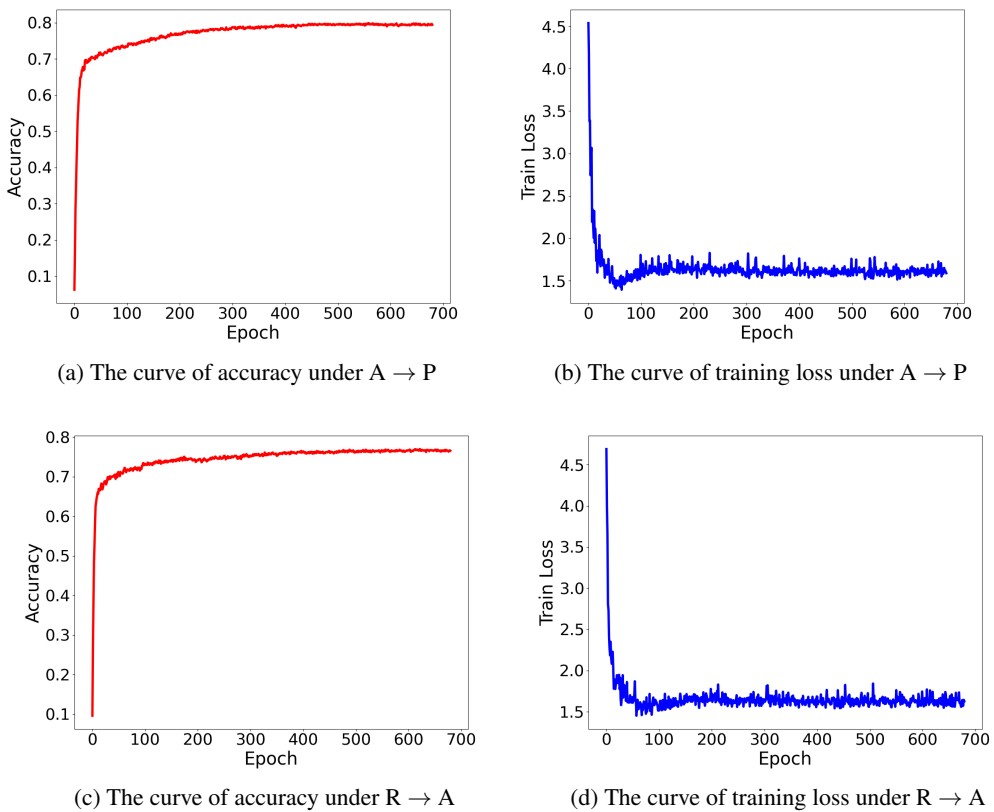

(a) The curve of accuracy under A → P

(b) The curve of training loss under A → P

(c) The curve of accuracy under R → A

(d) The curve of training loss under R → A

Figure 8: Convergence. The red curve is the changes of accuracy during the training process and the blue curve is the changes of training loss. The experiments are conducted on OfficeHome dataset, where (a) and (b) are in A → P setting, (c) and (d) are in R → A setting.

