# OpenReview forum: "SPA: A Graph Spectral Alignment Perspective for Domain Adaptation"
_NeurIPS.cc/2023/Conference — NeurIPS 2023 poster_

### Official Review · Reviewer_Ecsg · 2023-07-01

**Soundness:** 4 excellent
**Presentation:** 3 good
**Contribution:** 3 good
**Rating:** 7
**Confidence:** 5

**Summary:**

The authors present a new graph Spectral Alignment (SPA) framework for domain adaptation. The method consists of two main components: (i) a coarse graph alignment mechanism that uses a novel spectral regularizer to align domain graphs, and (ii) a fine-grained message propagation module that employs a neighbor-aware self-training mechanism to enhance discriminability in the target domain. Extensive experiments validate the effectiveness	of SPA.

**Strengths:**

The paper is exceptionally well-organized, with a clear and concise presentation of the proposed method. The approach itself is highly innovative and addresses a critical tradeoff in graph alignment. The authors have done an excellent job of explaining the core components of their method, including the coarse graph alignment mechanism and the fine-grained message propagation module.

**Weaknesses:**

I have a few concerns regarding some details in the paper that I would like to address:

1. I noticed that the Neighbor-aware Propagation is abbreviated as "L_nas" in the paper. However, I think that "L_nap" would be a more appropriate abbreviation.

2. In Table 4, where the results are presented without L_gsa and L_nas, the average accuracy is reported to be 68.5%. However, I am confused as to why this result is higher than the accuracy reported for CDAN in Table 2.

3. I am also curious about the performance of SPA when it is not combined with existing methods such as CDAN. Have you conducted any experiments to evaluate the single SPA's performance in isolation?

4. I am curious to know whether the graph in question is considered to be heavy. Additionally, I would like to inquire about the number of parameters that are present within the graph.

**Questions:**

See Weaknesses.

**Limitations:**

No limitations are included.

---

> ### Author Rebuttal · Authors · 2023-08-10
>
> Thank you for your constructive feedback, and we would like to address your concerns in detail below:
> > **[Weakness 1]** "[...] Neighbor-aware Propagation is abbreviated as "L_nas" in the paper. However, I think that "L_nap" would be a more appropriate abbreviation."
>
> **[Answer]** Thanks for your suggestion. We will modify this abbreviation for better presentation in the revised version.
>
>
> > **[Weakness 2]** "In Table 4, where the results are presented without L_gsa and L_nas, the average accuracy is reported to be 68.5%. However, I am confused as to why this result is higher than the accuracy reported for CDAN in Table 2."
>
> **[Answer]** In Table 2, following previous works [7, 12, 40], we compare the reported performance of CDAN [44] in the original paper. Besides, just as Line 182-184 state, we employ the standard cross-entropy loss with label-smoothing regularization [61] in our implementation. This label-smoothing regularization is directly applied to $L_{cls}$ , which is not related to $L_{gsa}$ or $L_{nas}$. In this way,  with this useful mechanism, even though the CDAN-based model is without $L_{gsa}$ and $L_{nas}$ in the first section of Table 4, the experimental results are better than the originally reported performance. In addition, following previous work [40], we also adopt coefficient warm-up trick during the training process, which also help improve the performance. More programming details can be referred in the code among our supplementary materials.
>
>
> > **[Weakness 3]** "I am also curious about the performance of SPA when it is not combined with existing methods such as CDAN. Have you conducted any experiments to evaluate the single SPA's performance in isolation?"
>
> **[Answer]** In Table 4, we point out the experiments of Ablation Study and Parameter Sensitivity are based CDAN [44]. Other ablation studies are default on DANN [17], which is just described as our Eq.(5) $L_{total}$.
> Here, we put more experimental details based on CDAN and DANN under the setting of OfficeHome dataset and we choose Gaussian similarity as the metric function.
> | OfficeHome        | A2C  | A2P  | A2R  | C2A  | C2P  | C2R  | P2A  | P2C  | P2R  | R2A  | R2C  | R2P  | avg  |
> | ----------------- | ---- | ---- | ---- | ---- | ---- | ---- | ---- | ---- | ---- | ---- | ---- | ---- | ---- |
> | CDAN w/ $L_{rwk}$ | 59.8 | 79.1 | 84.1 | 74.5 | 79.9 | 82.1 | 73.1 | 57.9 | 85.0 | 77.8 | 61.4 | 87.8 | 75.2 |
> | CDAN w/ $L_{sym}$	| 59.9 | 79.1 | 84.4 | 74.9 | 79.1 | 81.9 | 72.4 | 58.4 | 84.9 | 77.9 | 61.2 | 87.7 | 75.1 |
> | DANN w/ $L_{rwk}$	| 60.4 | 79.7 | 85.0 | 73.6 | 81.3 | 82.1 |	71.9 | 58.0 | 85.2 | 77.4 | 61.0 |	88.1 | 75.3 |
> | DANN w/ $L_{sym}$ | 60.0 | 79.8 | 84.8 | 73.5 | 80.8 | 82.5 | 72.0 | 57.5 | 84.4 | 76.9 | 60.7 | 87.77 | 75.1 |
>
> From these above experimental results, we can see that our performance works well without CDAN.
>
>
> > **[Weakness 4]** "I am curious to know whether the graph in question is considered to be heavy. Additionally, I would like to inquire about the number of parameters that are present within the graph."
>
> **[Answer]** Thanks for your thoughtful review. We can control the $k$-nearest neighbors for the graph, which is a hyper-parameter. Besides, during the graph construction process, we also can choose different metric function for edge weight. These experiments are offered in the Robust Analysis in experiments and appendix.

---

> > ### Author Response · Authors · 2023-08-18
> >
> > Dear Reviewer Ecsg,
> >
> > We have already heard back from Reviewer AnYf, Reviewer pjo2 and Reviewer Yk67. Do you have any further comments, concerns or questions towards our paper? Feel free to let us know. We are quite near the end of the discussion period :(
> >
> > -- The authors

---

> > ### Comment · Reviewer_Ecsg · 2023-08-21
> > **Official Commit of Submission13641 by Reviewer Ecsg**
> >
> > Thank you for answering my questions.
> > My problems are solved, and I think the paper should be accepted.

---

### Official Review · Reviewer_Yk67 · 2023-07-04

**Soundness:** 4 excellent
**Presentation:** 3 good
**Contribution:** 3 good
**Rating:** 6
**Confidence:** 4

**Summary:**

The paper proposes a domain adaptation method called Spectral Alignment (SPA). The proposed SPA utilizes spectral distance defined as the difference between eigenvalues of the Laplacian matrix to quantify the distribution gap. Then SPA employs this distance as regularizer for domain alignment. Additionally, SPA incorporates pseudo-labeling for feature aggregation, enhancing the similarity between semantically related nodes. The experimental results on multiple datasets verfy the effectiveness of the proposed algorithm.

**Strengths:**

Different from the previous domain adaptation methods, the paper introduces the spectral distance to measure the domain gap. In addition, the experiment results demonstrate the efficiacy of this idea.

**Weaknesses:**

1. The contribution of the paper is not clear. First, the paper directly employs a spectral distance from [1,20,59] without technical development to measure and narrow the distribution gap. Second, the feature aggregation with pseudo labels has been widely used in domain adaptation. So what the different of the proposed method to the previous work?
2. Figure 1 does not effectively demonstrate the motivation and innovative aspects of this paper.
3. Eq. (5) compute the total loss without consideration of loss scale. Specifically, Eq. (5) adopts four types of loss terms, L_{cls}，L_{adv}，L_{gsa}，L_{nas}. However, there is no trade-off parameters to balance different losses. I am not sure the convergence of the proposed method.


**Questions:**

1. The difference to the previous methods.
2. The convergence analysis of the proposed method.

**Limitations:**

Yes, the authors say that "The current method for constructing graph spectra is only a starting point and may be inadequate for more difficult scenarios such as universal domain adaptation. Additionally, this method is currently limited to visual classification tasks , and more sophisticated and generic methods to object detection or semantic segmentation are expected in the future" in line 324.

---

> ### Author Rebuttal · Authors · 2023-08-10
>
> Thank you for your constructive feedback, and we would like to address your concerns in detail below:
> > **[Weakness 1 + Question 1]** "The contribution of the paper is not clear. First, the paper directly employs a spectral distance from [1,20,59] without technical development to measure and narrow the distribution gap. Second, the feature aggregation with pseudo labels has been widely used in domain adaptation. So what the different of the proposed method to the previous work?"
>
> **[Answer]**
> Thanks for the feedback.
> First of all, the proposed definition of spectral distance is not the same one as the existing distances [1,20,59]. In practice, different metric functions and graph laplacians can lead to different model performance. More details of choosing different metric functions and graph laplacians can refer to our experiments.
>
> Besides, our dynamic graphs motivate us to adopt graph propagation method. We did not directly apply the classic propagation method to our model because of their computation cost. We optimized this classic method with batch-wise operations. We also did a lots of technical improvement to joint the graph spectral alignment and neighbor-aware propagation and pursue a balance between the transferability and disciminability of model.
>
>
> > **[Weakness 2]** Figure 1 does not effectively demonstrate the motivation and innovative aspects of this paper.
>
> **[Answer]** Thanks for your thoughtful suggestion. This figure mainly used to illustrate the constitution of our framework. We will polish this pipeline to highlight our motivation and novelty in the revised version.
>
> >  **[Weakness 3 + Question 2]** "Eq. (5) compute the total loss without consideration of loss scale. Specifically, Eq. (5) adopts four types of loss terms, L_{cls}，L_{adv}，L_{gsa}，L_{nas}. However, there is no trade-off parameters to balance different losses. I am not sure the convergence of the proposed method. The convergence analysis of the proposed method."
>
> **[Answer]** Thanks for your pointing out this problem. In our implementation, we indeed use loss scales. In Eq. (3), we also write out $\alpha$, which is the coefficient term properly designed to grow along with iterations to mitigate the noises in the pseudo-labels at early iterations and avoid the error accumulation. In the revised version, we will fix the scale coefficients for other loss terms in Eq. (5).
>
> To analyze the loss scales of SPA, here are more experiments of sensitivity analysis based on DANN [17] under the setting of OfficeHome dataset and we choose Gaussian similarity as the metric function.
> The experimental results are shown in the following table.
>
> Fixing the coefficient of $L_{nas}$ = 0.2, the coefficient of $L_{gsa}$ changes from 0.1 to 0.9.
>
> | The coefficient of $L_{gsa}$ = | 0.1  | 0.3  | 0.5  | 0.7  | 0.9  | $\Delta$ |
> | ------- | ---- | ---- | ---- | ---- | ---- | -------- |
> | A2C     | 60.0 | 59.8	| 60.3 | 60.1 | 59.0 | 1.3      |
> | C2R     | 82.4 | 82.1	| 82.3 | 81.8 | 82.0 | 0.6      |
>
> Fixing the coefficient of $L_{gsa}$ = 1.0, the coefficient of $L_{nas}$ changes from 0.1 to 0.9.
>
> | The coefficient of $L_{nas}$ = | 0.1  | 0.3  | 0.5  | 0.7  | 0.9  | $\Delta$ |
> | ------- | ---- | ---- | ---- | ---- | ---- | -------- |
> | A2C     | 59.0 | 60.4 | 59.4 | 58.9 | 58.3 | 2.1      |
> | C2R     | 81.0 | 82.1	| 82.8 | 83.4 | 83.2 | 2.4      |
>
> From the series of results,
> we can find that in OfficeHome dataset, the choice of different coefficients result in similar results,
> which means that SPA is insensitive to these coefficients.
>
> Beside, we also visualize the curve of training loss and accuracy to support the convergence of our method and we put these figures in .pdf file.
>
> Hope these extra experiments and analysis can address your concerns.

---

> > ### Comment · Reviewer_Yk67 · 2023-08-17
> >
> > Thanks for your reply and your efforts in the additional experiments. However, I would not change my score since the author did not provide theoretical analysis to illustrate the differences between the algorithm in this article and previous work.

---

> > > ### Author Response · Authors · 2023-08-17
> > >
> > > We thank the reviewer for the reply. As as stated in both the paper as well as the first reply, indeed that the theoretical analysis of spectral graph alignment are not presented. However, we argue that because graph spectra and graph matching are both missing--- not only in our paper, but also in the entire field of graph matching ---so that making the theory advancement is difficult and certainly beyond the scope of this paper.
> > >
> > > We thank the reviewer for pointing out the related works above. Despite, we went through all these papers [1, 20, 59], and we believe they mostly belongs to a different field of mathmatical sciences. By contrast, when we see the DA papers that being published in recent years [7, 13, 30, 47, 80, 82], very few paper factually approached the problem as a graph matching problem from the theoretical aspect. Currently, the entire field of graph matching puts more attention on optimal solutions, remaining a relatively blank understanding towards learning and generalization.
> > >
> > > Put these points together, we believe it is more suitable for a set of future works towards resolving your issue. Nevertheless, we are still immensely greatful for the good words and positivity the reviewer has exposed.

---

### Official Review · Reviewer_pjo2 · 2023-07-04

**Soundness:** 2 fair
**Presentation:** 2 fair
**Contribution:** 3 good
**Rating:** 6
**Confidence:** 4

**Summary:**

The paper proposed a graph SPectral Alignment (SPA) framework which uses a coarse graph spectral alignment and neighbor-aware self-training mechanism. The authors compared the proposed method with different UDA methods using several benchmark datasets.

**Strengths:**

The paper is generally well-written and clear. The results show the superior performance of the method in terms of accuracy in different datasets.

**Weaknesses:**

however, it missed some details that need to be completed.

**Questions:**

1. Please add more details that how to construct a dynamic graph, especially for weighted edge and metric function.
2. Please explain the reason why decreasing the distance of graphs is useful for solving da problem.
3. Line 121, ``with n vertices can be from the spectrum of any other graph with n vertices". what's the meaning of n? Do both graphs have to require n?
4. Why uses weighed k-Nearest-Neighbor classification to obtain the pseudo-label？
5. Please add more analysis about the results of experiments, such as Line 226, ``The experiments shows SPA consistently ourperforms than various of DA methods".

---

> ### Author Rebuttal · Authors · 2023-08-10
>
> Thanks sincerely for the constructive reviews, and we have made great efforts to address all these concerns.
> >  **[Weakness 1]** "Please add more details that how to construct a dynamic graph, especially for weighted edge and metric function."
>
> **[Answer]** Thanks for your suggestion. We put different choices of metric function in the robust analysis in experiments and leave out some repeated information because of page limits. In revised version, we will follow your suggestion to add more details and analysis in Section 3.1 and experiments.
>
>
> >  **[Weakness 2]** Please explain the reason why decreasing the distance of graphs is useful for solving da problem.
>
> **[Answer]** Thanks for your suggestion.
> The final goal of domain adaptation is to learn the domain-invariant representations (Line 20). In our paper, we first construct dynamic graphs within source domain and target domain separately and then utilize a spectral distance to project these graphs into the spectral space. After spectral alignment, source domain graphs are closer to target domain graphs in the latent space (Line 144-145). With message propagation, this alignment is further encouraged. In this way, the whole framework can perform well on domain adaptation scenario.
>
>
> >  **[Weakness 3]** Line 121, ``with n vertices can be from the spectrum of any other graph with n vertices". what's the meaning of n? Do both graphs have to require n?
>
> **[Answer]**
> Thanks for your question.
> In our paper, the node size 'n' is exactly the batch size, which is a hyper-parameter.
> Under our setting, following the definition of a classic graph matching problem, source domain graphs always have the same 'n' as target domain graphs. Graph matching aims at finding the vertex correspondence between the two unlabeled graphs that maximizes the total edge weight correlation. In other words, we expect two randomly sampled subsets from source/target domains shares similar intrinsical topological properties in the latent space and thus, have great transferability.
> There are also some papers that study inexact graph matching, which allows matching two graphs of different sizes. We leave the extension to inexact graph matching for future work.
>
> >  **[Weakness 4]** "Why uses weighed k-Nearest-Neighbor classification to obtain the pseudo-label？"
>
> **[Answer]**
> Thanks for your question. We'd like to show the superiority of our kNN pseudo-labeling algorithm from two aspects.
>
> **Motivation**: we can regard the weighted kNN method as a type of regularization. Our source domain graph and target domain graph are within a domain and follow homophily assumption. Compared with the single-point pseudo-labeling method, we believe the kNN method is able to encourage the smoothness of predictions among neighbors and hence better performance.
>
> **Technique**:
> after graph spectral alignment, the rich structure information is coarsely transferred to the target domain and then we hope to align those fine-grained intra-domain information (Line 146-147). Message propagation is encouraged to perform this alignment.
>
> The classic label propagation method can be represented as $\mathbf{Z} = (\mathbf{I} - \alpha \mathbf{A})^{-1}$ $\mathbf{Y}$. The rows of $\mathbf{Y}$ corresponding to labeled examples are one-hot encoded labels and the rest are zero. The $\alpha \in [0,1)$ is a parameter. The class prediction for an unlabeled example $x_i$ is $\hat{y} = \arg \max_{j} z_{i, j}$, where $z_{i, j}$ is the $(i, j)$ element of matrix $\mathbf{Z}$.
>
> However, the computation of propagation matrix $\mathbf{Z}$ is usually performed on all data samples, and implicitly requires access to all embedding vectors at each step of computation.
> Therefore, we focus on batch-wise operations and avoid to solve the above linear system of message propagation via the weighted KNN classification algorithm [72] to generate pseudo-labels for target domain graphs.
>
>
> >  **[Weakness 5]** "Please add more analysis about the results of experiments, such as Line 226, 'The experiments shows SPA consistently outperforms than various of DA methods'."
>
> **[Answer]**
> Thanks for your suggestion. We leave out some information because of page limits and more analysis of experimental results can be found in appendix. In revised version, we will follow your suggestion to add more details and analysis for better presentation.

---

> > ### Comment · Reviewer_pjo2 · 2023-08-18
> >
> > I thank the authors for the response and have changed the rating score.

---

> > > ### Author Response · Authors · 2023-08-18
> > >
> > > Thank you for the support!

---

### Official Review · Reviewer_AnYf · 2023-07-06

**Soundness:** 3 good
**Presentation:** 4 excellent
**Contribution:** 4 excellent
**Rating:** 8
**Confidence:** 5

**Summary:**

This paper proposes a novel graph spectral alignment framework for unsupervised domain adaptation, jointly balancing the inter-domain transferability and the intra-domain discriminability. First, they cast the domain adaptation problem to graph primitives by composing a coarse graph alignment and aligning domain graphs in eigenspaces. Second, they perform a fine-grained message passing in the target domain via a neighbor-aware self-training mechanism. The first alignment method gives rise to coarse-grained topological structures transfer across domains but in a more intrinsic way than restrictive point-wise matching. With the help of fine-grained neighbor-aware propagation, the whole framework is able to refine the transferred topological structure to produce a discriminative domain classifier.

**Strengths:**

1.	This paper is easy to follow with clear writing, concise figures, and well-structured formulation in problem definition and methods.
2.	The research problem is essential and motivation is clear. This paper gives a new perspective for the essential problem of how to find a suitable utilization of intra-domain information and inter-domain information in unsupervised domain adaptation.
3.	The proposed graph spectral alignment method is an attractive method in domain adaptation, which works well and inspires follow-up research.
4.	The experiment part is comprehensive, including 4 commonly used datasets in domain adaptation.


**Weaknesses:**

1.	To support the claim of balancing transferability and discriminability, the authors offer the figures of visualized features. More ablation studies to express the balance can be offered.
2.	The introduction of this paper is less informative. More technical details of the proposed method can be provided there to improve legibility, as well as more comparative description of previous works.
3.	Some grammars need to be improved. Line 120-122, the long sentence with two predicates. Line 241, ‘achieve’ should add ‘s’. Line 265, ‘presents’ should remove ‘s’. Line 270, ‘also use’ should add ‘s’. Line 283, ‘, By’ should use period. Line 289-292, ‘propose’, ‘formulate’, ‘calculate’ should add ‘s’. Line 297, ‘comes out’ should remove ‘s’.


**Questions:**

1.	The questions are mentioned in weakness. More ablation studies can be offered. More information can be added to introduction.
2.	For related work part, more latest references about graph methods can be added.
3.	For the Laplacian metrices, you report the results based on two normalized types. Did you try any other types?
4.	The authors mentioned the relation between kNN pseudo-labeling and graph propagation, which is confusing for me. Can you explain more on this?


**Limitations:**

Yes, the authors have addressed the limitations of their work at the end of the paper.

---

> ### Author Rebuttal · Authors · 2023-08-10
>
> Thank you for these thoughtful comments, and we would like to address your concerns in detail below:
> > **[Weakness 1 + Question 1]** "To support the claim of balancing transferability and discriminability, the authors offer the figures of visualized features. More ablation studies to express the balance can be offered."
>
> **[Answer]** Thank you for expressing your concern regarding the balance between transferability and discriminability of our method. We offer feature visualizations of C->R setting of OfficeHome dataset in the main paper. More visualized features of A->D, A->W setting of Office31 dataset and A->C setting of OfficeHome dataset in the appendix.
>
> Besides, we offer a Transferability and Discriminability section in appendix. We use A-distance [27] measures the distribution discrepancy, which shows our SPA model achieves a lower generalization error.
>
> Furthermore, following previous work [3], we also offer the source accuracy and target accuracy specifically in Figure 2b and Figure 2c in appendix, which illustrates our SPA can always achieve higher target accuracy. Combined with all of these experimental results, it reveals that our SPA enhances transferability while still keep a strong discriminability.
> Hope this can address your concern.
>
>
> > **[Weakness 2 + Questions 1]** "The introduction of this paper is less informative. More technical details of the proposed method can be provided there to improve legibility, as well as more comparative description of previous works. [...] More information can be added to introduction."
>
> **[Answer]** Thanks for your suggestion. We leave out some information because of page limits. More comparisons with previous works are in related work. We will add more details and highlights to the introduction in the revised revision as the reviewer suggests.
>
>
> > **[Weakness 3]** "Some grammars need to be improved. Line 120-122, [...]"
>
> **[Answer]** Thank you for pointing out these grammar problems. We will fix these grammars for more clear presentation in the revised revision.
>
>
> > **[Question 2]** For related work part, more latest references about graph methods can be added.
>
> **[Answer]** Thank you for your suggestion. Although our method focuses mainly on graph matching and alignment methods for domain adaptation, we will improve our reference with more papers about graph data mining methods or graph neural networks for domain adaptation as suggested by the reviewer.
>
>
> > **[Qeustion 3]** For the Laplacian matrices, you report the results based on two normalized types. Did you try any other types?
>
> **[Answer]** Thanks for your question. We choose random walk Laplacian matrix and symmetrically normalized Laplacian matrix for robustness analysis in experiment section. These two types of graph laplacians are commonly-used among graph methods, because they have very good properties such as being symmetric, positive semi-definite, and having non-negative eigenvalues, which is an important basis for following spectral alignment.
>
> Furthermore, here we offer some extra experimental results for a type of unnormalized laplacian matrix $L = D - A$ with cosine similarity and Guassian similarity respectively in the following:
> | OfficeHome       | A2C  | A2P  | A2R  | C2A  | C2P  | C2R  | P2A  | P2C  | P2R  | R2A  | R2C  | R2P  | avg  |
> | ---------------- | ---- | ---- | ---- | ---- | ---- | ---- | ---- | ---- | ---- | ---- | ---- | ---- | ---- |
> | w/ $cos$  | 42.1 | 67.9 | 71.9 | 38.8 | 51.2 | 55.7 | 55.8 | 32.4 | 71.9 | 66.1 | 48.7 | 80.2 | 56.9 |
> | w/ $gas$  | 50.2 | 65.7 | 75.4 | 56.3 | 53.3 | 61.1 | 63.2 | 44.7 | 66.8 | 64.2 | 47.5 | 72.1 | 60.1 |
>
> From these experimental results, it also illustrates that the normalized graph laplacians with good properties perform better than the unnormalized one.
>
>
> > **[Question 4]** The authors mentioned the relation between kNN pseudo-labeling and graph propagation, which is confusing for me. Can you explain more on this?
>
> **[Answer]** Thanks for your thoughtful review.
> For graph propagation methods, it can be represented as $\mathbf{Z} = (\mathbf{I} - \alpha \mathbf{A})^{-1}$ $\mathbf{Y}$. The rows of $\mathbf{Y}$ corresponding to labeled examples are one-hot encoded labels and the rest are zeros. The $\alpha \in [0,1)$ is a parameter. The class prediction for an unlabeled example $x_i$ is $\hat{y} = \arg \max_{j} z_{i, j}$, where $z_{i, j}$ is the $(i, j)$ element of matrix $\mathbf{Z}$.
> Based on our constructed graphs, it is intuitive for us to adopt graph propagation.
> However, the computation of $\mathbf{Z}$ is performed on all data samples, and implicitly requires access to all embedding vectors at each step of computation.
> If the size of dataset increases, it would be intractable to finish the computation of matrix $\mathbf{Z}$.
> We focus on producing labels for unlabeled target domain graphs in each iteration.
> Therefore, we avoid to solve the above linear system of classic label propagation via the weighted KNN classification algorithm [72] to generate pseudo-labels for target domain graph.

---

> > ### Comment · Reviewer_AnYf · 2023-08-17
> >
> > Thanks for your reply.

---

### Official Review · Reviewer_54MB · 2023-07-07

**Soundness:** 2 fair
**Presentation:** 3 good
**Contribution:** 2 fair
**Rating:** 3
**Confidence:** 5

**Summary:**

This work proposed a graph spectral alignment model for unsupervised domain adaptation. They evaluated on several benchmarks and show good performance, especially in DomainNet and Office-Home, compared with existing UDA methods. Generally, the paper is easy to follow.

**Strengths:**

The paper is generally easy to follow and the experimental results on DomainNet and Office-Home are pretty good.

**Weaknesses:**

The novelty of this work is limited. The graph alignment is simply extended from BSP [12] and Graph Matching like SIGMA [39]. The Neighbor-aware Self-training Mechanism is similar to embedding propagation. In this sense, there is limited novelty in terms of the methodology contribution.

The performance on other benchmarks like Office-31 and VisDA seem not better than others. The visualization of feature embedding is not informative enough to validate the graph alignment. It is essential to visualize the domain-wise graphs, and how are they aligned? This can show more insights. For example, in Office-Home, their model performs better when A is target domain. Then what is the key reason?



**Questions:**

What are the cross-domain graphs? How are they aligned?

What is key difference in terms of the proposed two terms?



**Limitations:**

The experimntal results are not diverse, mainly on classification performance.

The model is incremental over existing methods.

---

> ### Author Rebuttal · Authors · 2023-08-10
>
> Thank you for your valuable time and review, and we have made great efforts to address all these concerns below:
> > **[Weakness 1]** "The novelty of this work is limited. The graph alignment is simply extended from BSP [12] and Graph Matching like SIGMA [39]. The Neighbor-aware Self-training Mechanism is similar to embedding propagation. In this sense, there is limited novelty in terms of the methodology contribution."
>
> **[Answer]** Thanks for the feedback.
> As we stated in the (Line 295-297 and Line 316), both work ([9] and [39]) are indeed related to our work.
> The key finding of BSP [9] is that the eigenvectors with the largest singular values will dominate the feature transferability and thus, BSP penalizes the largest singular values for improving transferability.
> SIGMA [39] focuses on the category-level adaptation of object detection. Technically, their bipartite graph matching is a type of exact graph matching, requiring multiple matching stages for nodes and edges respectively.
>
> Despite that, as is stated by multiple previously published papers [9, 12, 32], the very core and crucial challenge of DA (and UDA) attributes to a balancing between inter-domain transferability and intra-domain discriminability (in our paper: Line 23-27).
> To approach this balancing problem is not trivial since the remarkable transferability is usually enhanced at the expense of worse discriminability (in our paper: Line 30-31).
>
> Our work's major finding centers at this core problem as we develop a joint adjusting mechanism of graph spectral alignment and message propagation module.
> To make the graph propagation feasible in UDA, we present a novel batch-wise propagation operator and develop the self-training propagation method, which is not a direct application of existing method.
>
> In comparison with the two works the reviewer mentioned, while we do believe they make substantial contributions to the community, we believe neither work concerns this essential balancing problem.
>
>
>
> > **[Weakness 2]** "The performance on other benchmarks like Office-31 and VisDA seem not better than others. "
>
> **[Answer]**
> First, while the reviewer (rightfully) point this out, we may still want to list the performance gain of our approach overall: +2.6% on OfficeHome, +0.5% on VisDA, neck-to-neck performance with current SOTA[48] on Office-31, and **most notably*** the large-scale DomainNet with +8.6%.
> Partially, the reviewer also gave us credits on the empirical performances as is written in Strength section of th review :)
>
> Upon that, we pledge the reviewer to reconsider the empirical aspect of our approach, especially crediting the significant performance plus on the large-scaled one (DomainNet).
> To have this result was somewhat rare, as we posit that a certain amount of prior papers did not report original DomainNet in the experiments.
>
>
>
> > **[Weakness 3 + Questions]** "The visualization of feature embedding is not informative enough to validate the graph alignment. It is essential to visualize the domain-wise graphs. What are the cross-domain graphs? How are they aligned? What is key difference in terms of the proposed two terms?"
>
> **[Answer]**
> There seems some misunderstanding here. The graphs we built in our approach always attribute to each individual domain. As such, we do not have the domain-wise or the cross-domain graphs in that sense.
> We kindly ask the reviewer to reiterate Line 96-104 for the graph definition, as well as the visualization plots in both experiments and the appendix.
>
>
>
> > **[Weakness 4]** "For example, in Office-Home, their model performs better when A is target domain. Then what is the key reason?"
>
> **[Answer]**
> This is a good observation. As we can see from Table 2, the performances across different setups do have exhibited some patterns: C2A > A2C, A2P > P2A, A2R > R2A, as is rightfully pointed out by the reviewer.
> However, all the baselines in Table 2 also exhibit the same pattern --- not just our approach.
> We postulate that this very much aligned results reveal a intrinsic properties in these public datasets --- and perhaps more importantly --- a difference of difficulty-level for these different pair of domain transfer.
>
>
>
> > **[Limitations]** "The experimental results are not diverse, mainly on classification performance."
>
> **[Answer]**
> As we dive into the literature, most papers published in the past ML venues (such as ICML, ICLR and NeurIPS) on DA, DG or UDA [9, 12, 17, 18, 31, 32, 44, 48], have commonly stuck with the benchmarks we report in our paper. We believe this is due to that this line of work mostly focuses on the fundamental ML problem behind UDA, rather than specific task.
> On the other line, we do acknowledge that there are papers covering other diversified tasks such as object detection [39], semantic segmentation [36], or NLP [8, 10]. Different from the ML line of work, these papers were dedicated at specific task rather than promoting the generality of the method.
>
> That said, given the terrain of the normal empirical proxy of UDA, we believe our experiments suffice to establish our contribution.

---

> > ### Author Response · Authors · 2023-08-18
> >
> > Dear Reviewer 54MB,
> >
> > Hi! Since the beginning of the discussion, we have already heard back from three other reviewers (AnYf, pjo2, Yk67).
> >
> > And luckily, it seems that all the other reviewers have reached a consensus of positivity towards our paper :)
> >
> > We may humbly ask the reviewer if there are any other questions, concerns, comments or responses that the reviewer may have. Feel free to post them while we can still clarify and help, before this discussion phase ends (and it will end soon :(
> >
> > Thank you!
> >
> > -- The authors

---

### Author Rebuttal · Authors · 2023-08-10

We thank the reviews for their positive views about different aspects of our work which say that:
* our proposed method is: "an attractive method which works well and inspires follow-up research" (Reviewer AnYf), "different from the previous domain adaptation methods" (Reviewer Yk67), "highly innovative" (Reviewer Ecsg)
* our motivation is: "essential and clear" (Reviewer AnYf), "addresses a critical trade-off in graph alignment" (Reviewer Ecsg)
* our formulation is: "well-structured formulation in problem definition and methods" (Review AnYf), "an excellent job" (Reviewer Ecsg)
* our presentation is: "easy to follow" (Reviewer 54MB, Reviewer AnYf), "well-written and clear" (Reviewer pjo2), "clear writing, concise figures" (Review AnYf), "a clear and concise presentation of the proposed method" (Review Ecsg)
* our experiment is: "pretty good" (Reviewer 54MB), "comprehensive" (Reviewer AnYf), "superior" (Reviewer pjo2), "the experiment results demonstrate the efficacy of this idea" (Reviewer Yk67)

We also thank the reviewers for all the other detailed and insightful comments. We would like to address their concerns in detail below.

---

### Comment · Area_Chair_jo9g · 2023-08-20
**Kindly remind reviewers to evaluate authors' responses**

Dear Reviewers,

Hope this message finds you well. We appreciate your dedicated reviews for NeurIPS 2023.
Kindly note that authors have responded to your feedback. Your prompt evaluation of their responses is crucial for finalizing papers.
Please log in and assess their responses at your earliest convenience. Your timely input ensures paper improvements and conference quality.
Access the system to review responses. Reach out if you need assistance.

Reviewer 54MB currently has a different opinion with the other reviewers. Please provide your comments on the authors' responses in time.

Best regards,

Area Chair

---

### Decision · Program_Chairs · 2023-09-21

**Decision:**

Accept (poster)

**Comment:**

This paper introduces the Graph Spectral Alignment (SPA) framework, which effectively balances inter-domain transferability and intra-domain structures, outperforming the state-of-the-art domain adaptation methods on the standardized benchmarks. It is well-written and structured, providing a fresh perspective on the challenge of balancing intra-domain and inter-domain information in unsupervised domain adaptation, with an attractive and effective method, supported by the comprehensive experiments. However, the paper also lacks novelty in methodology, shows mixed performance results on certain benchmarks, and lacks informative feature visualization. In the rebuttal, the authors thoroughly addressed reviewers' questions and supplemented the relevant results. While there are varying opinions among the reviewers, most agreed to accept the revised paper. Despite some differing views, the AC still considers this paper to be an excellent one and therefore decides to accept it.